# HDLBP binds ER-targeted mRNAs by multivalent interactions to promote protein synthesis of transmembrane and secreted proteins

Ulrike Zinnall [1,8], Miha Milek [1,2,7,8 ✉], Igor Minia[1], Carlos H. Vieira-Vieira [1], Simon Müller[3], Guido Mastrobuoni [1], Orsalia-Georgia Hazapis[1], Simone Del Giudice [1], David Schwefel [4], Nadine Bley[3], Franka Voigt[5], Jeffrey A. Chao [5], Stefan Kempa [1], Stefan Hüttelmaier [3], Matthias Selbach [1,4] & Markus Landthaler [1,6 ✉]

The biological role of RNA-binding proteins in the secretory pathway is not well established. Here, we describe that human HDLBP/Vigilin directly interacts with more than 80% of ER-localized mRNAs. PAR-CLIP analysis reveals that these transcripts represent high affinity HDLBP substrates and are specifically bound in their coding sequences (CDS), in contrast to CDS/3'UTR-bound cytosolic mRNAs. HDLBP crosslinks strongly to long CU-rich motifs, which frequently reside in CDS of ER-localized mRNAs and result in high affinity multivalent interactions. In addition to HDLBP-ncRNA interactome, quantification of HDLBP-proximal proteome confirms association with components of the translational apparatus and the signal recognition particle. Absence of HDLBP results in decreased translation efficiency of HDLBP target mRNAs, impaired protein synthesis and secretion in model cell lines, as well as decreased tumor growth in a lung cancer mouse model. These results highlight a general function for HDLBP in the translation of ER-localized mRNAs and its relevance for tumor progression.

---

[1] Max Delbrück Center for Molecular Medicine in the Helmholtz Association (MDC), Berlin Institute for Medical Systems Biology, Berlin, Germany. [2] National Institute of Chemistry, Ljubljana, Slovenia. [3] Institute of Molecular Medicine, Medical Faculty, Martin Luther University, Halle, Germany. [4] Charite-Universitätsmedizin, corporate member of Freie Universität Berlin and Humboldt-Universität zu Berlin, Institute of Medical Physics and Biophysics, Berlin, Germany. [5] Friedrich Miescher Institute for Biomedical Research, 4058 Basel, Switzerland. [6] IRI Life Sciences, Institute of Biology, Humboldt-Universität zu Berlin, Berlin, Germany. [7] Present address: Core Unit Bioinformatics, Berlin Institute of Health at Charité, Berlin, Germany. [8] These authors contributed equally: Ulrike Zinnall, Miha Milek. ✉ email: miha.milek@bih-charite.de; markus.landthaler@mdc-berlin.de

In eukaryotic cells, the localization of functional protein products is largely determined by the site of their translation. While soluble proteins are translated in the cytosol, co-translational targeting to the endoplasmic reticulum (ER) enables newly synthesized proteins to enter the secretory pathway, resulting in their secretion or membrane integration[1–4]. The canonical secretory pathway initiates in the cytosol with the synthesis of the hydrophobic targeting signal (signal peptide or transmembrane domain)[5]. Subsequent binding of the signal recognition particle (SRP) to the nascent peptide results in ribosome elongation arrest and formation of the ribosome nascent chain complexes (RNCs)[6]. This allows the re-localization of the cytosolic SRP-RNC to the ER membrane via the SRP receptor and translocation of the nascent peptide to the ER lumen[7]. In recent years, a non-canonical SRP-independent pathway was discovered in yeast[8] along with the evidence for the recruitment of the SRP to the mRNA prior to ribosome engagement[9] and SRP-independent ER targeting[10–12]. This raises the possibility of the existence of yet unknown mechanisms for the recognition of membrane-bound mRNAs.

The potential role of regulatory elements in mRNA sequences for ribosome elongation arrest and nascent chain recognition are poorly understood. Several studies have identified elements within coding sequences (CDS) and 3' untranslated regions (3' UTRs) that may distinguish ER-bound from cytosolic mRNAs[13–17]. However, trans-acting factors that may be responsible for the recognition of such elements are unknown. Recently, it was observed that a small subset of mRNAs that encode soluble proteins may also be localized and translated at the ER, indicating additional mechanisms that regulate the fate of a localized mRNA[18]. While there is evidence for subpools of ER-associated ribosomes that interact with pyruvate kinase in muscle[19], comprehensive differences in the composition, assembly and active translation states of cytosolic and ER-bound ribosomes have not been identified[20–22]. Furthermore, a previously unknown variant of the ribosome-dependent nonsense-mediated decay (NMD) pathway was discovered at the ER, hinting at an additional layer of regulation for ER-bound mRNAs[23]. In summary, the translational fate of mRNAs encoding soluble and membrane proteins may be tightly regulated by trans-acting factors such as RNA-binding proteins, which could function beyond the canonical SRP-dependent model.

HDLBP (also known as VIGILIN) is a conserved and ubiquitously expressed RBP localized both to the cytosol and the ER membrane[24,25]. It contains 15 hnRNPK-homology (KH) RNA-binding domains (RBDs)[26]. KH domains are high-affinity RNA-recognition elements (RREs), most commonly tetranucleotides as observed for FMRP[27], SF1, HNRNPK, and others[28]. Some may also recognize bipartite motifs, e.g., IGF2BP protein family[29–32]. HDLBP and its yeast orthologue SCP160 have been found to contribute to many biological processes[33] such as translation[34,35] or protein aggregation[36], and have been linked to carcinogenesis[37,38]. Recently, HDLBP has been shown to be required for replication of flaviviruses ZIKV and DENV, most likely by increasing the translation efficiency of viral proteins at the ER[24]. HDLBP is also a promising target for cardiovascular research, since it promotes secretion of very-low-density lipoprotein (VLDL) and leads to less atherosclerotic plaques upon hepatic HDLBP knockdown in atherosclerosis prone Ldlr$^{-/-}$ mice[35]. It was proposed that HDLBP binds to CHHC or CHYC (H = A/C/U and Y = C/U) containing regions in a subset of mRNAs encoding for secreted proteins and enhances their translation. However, functional aspects of HDLBP binding to RNA and mechanistic events during translation remain uncertain.

Here, we assayed HDLBP binding sites in a transcriptome-wide manner by PAR-CLIP and discovered their potential function as selective sequence determinants of ER-bound mRNAs. HDLBP directly and specifically interacted with more than 80% of all ER-localized mRNAs and was primarily bound to long CU-rich motifs in their CDS, a unique feature which is much more frequently found in membrane-bound compared to cytosolic mRNAs. Biochemical, transcriptomic, and proteomic methods were used to evaluate the functional consequences of HDLBP absence on ER translational efficiency, protein synthesis, and secretion and highlighted its requirement for these biological processes. Finally, we expanded our findings to an in vivo system and evaluated the effect of the absence of HDLBP on tumor formation capacity.

## Results

**HDLBP directly interacts with ER-targeted mRNAs.** To functionally characterize the interactions of HDLBP with the localized transcriptome, we quantified cytosolic and membrane-bound mRNAs in HEK293 cells. Using a subcellular chemical fractionation approach[39], we obtained cytosol and membrane fractions, as evidenced by the presence or absence of compartment-specific protein markers (Supplementary Fig. 1a). To obtain localized mRNA profiles, we next performed mRNA-seq from whole-cell, cytosol, and membrane fractions (Supplementary Fig. 1b), resulting in highly reproducible mRNA abundance quantifications (Supplementary Fig. 1c). Due to the bimodal distribution of membrane-to-cytosol enrichment (Fig. 1a), we were able to classify over 7000 mRNAs according to their partitioning between the cytosol and membrane (Supplemental Data 1). To validate this result, we compared the enrichment values between mRNAs that encoded or lacked co-translational targeting signals. As expected, we found that mitochondrial DNA-encoded proteins, as well as signal peptide (SP) and transmembrane helices (TM) encoding mRNAs had much higher membrane enrichment when compared to mRNAs encoding post-translationally targeted tail-anchored proteins or nuclear DNA-encoded mitochondrial proteins (Fig. 1b), validating our approach.

Successful quantification of mRNA localization prompted us to determine the whole-cell HDLBP-bound transcriptome in HEK293 cells by PAR-CLIP (Supplementary Fig. 1d, e). The majority of FLAG/HA-HDLBP cross-linking signal, detected as T-C transitions, was present in coding sequences (CDS) and 3' untranslated regions (3' UTRs) of mRNAs (Supplementary Data 2, 3), as well as in rRNA and tRNAs (Fig. 1c). After expression-level normalization (Supplementary Fig. 1f, g), the PAR-CLIP T-C signal of membrane-bound mRNAs was normally distributed without mRNA abundance bias (Fig. 1d), suggesting high specificity of HDLBP interactions for this pool of mRNAs. Interestingly, high HDLBP PAR-CLIP enrichment in membrane-localized mRNAs was mostly due to high CDS interactions rather than 3' UTR binding (Supplementary Fig. 1h). We estimated that more than 80% of the total membrane-bound mRNA pool was bound by HDLBP (Fig. 1e). The unbound membrane-localized mRNAs included 13 mitochondrial DNA-encoded genes (Supplementary Fig. 1i), which were not expected to be HDLBP targets since HDLBP is not localized to mitochondria[25]. Correspondingly, HDLBP-bound mRNAs were highly enriched for SP and TM helix encoding transcripts, while a significantly weaker binding was observed for cytosolic mRNAs and those encoding mitochondrial proteins (Fig. 1f). In addition, we found that mRNAs encoding tail-anchored membrane proteins were not enriched, confirming that HDLBP binds to the endoplasmic reticulum (ER)-targeted mRNAs with high specificity.

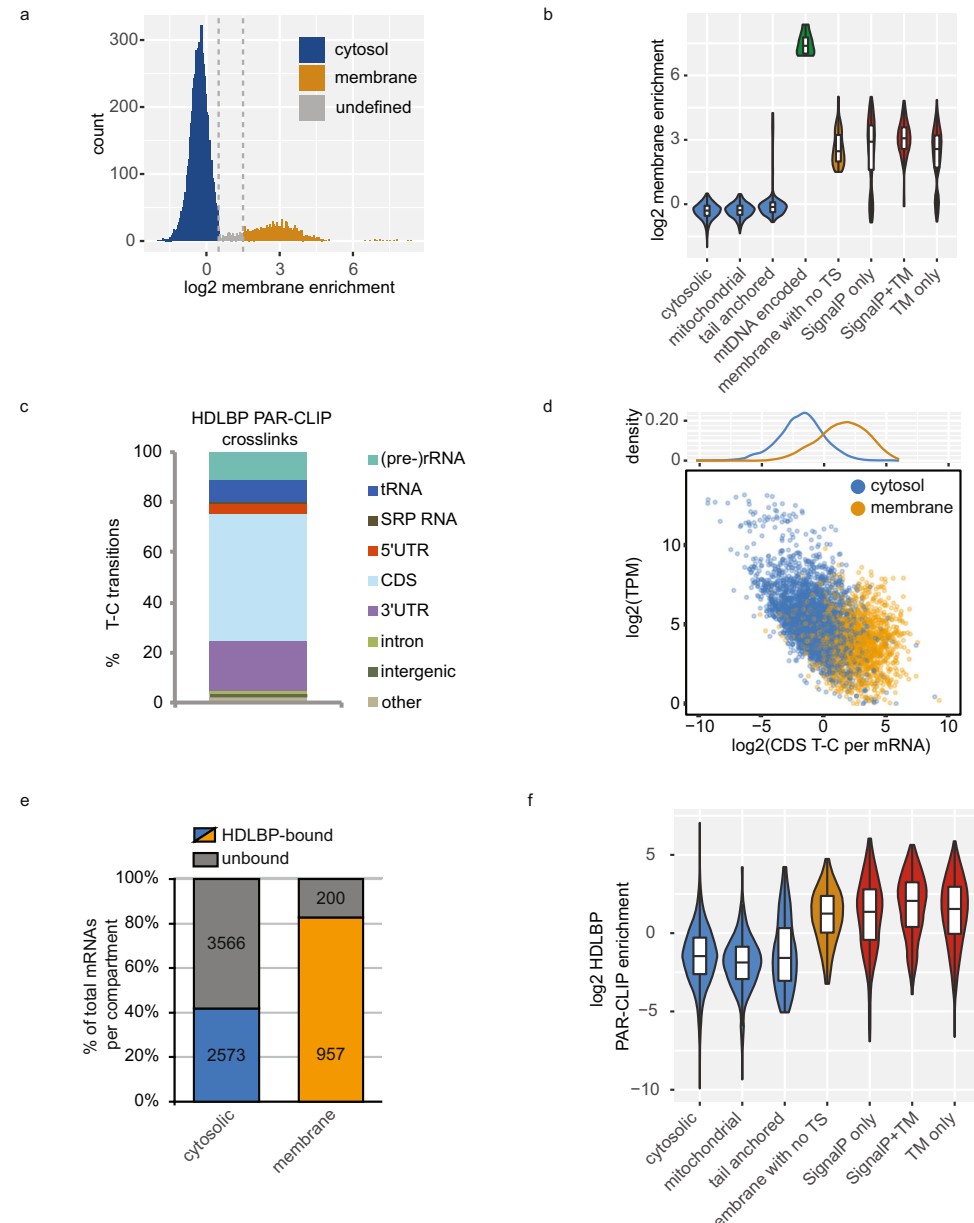

**Differential HDLBP binding to transcript regions of membrane-bound and cytosolic mRNAs**. To determine the molecular characteristics of HDLBP–RNA interactions, we first examined the relative distribution of HDLBP cross-linking positions within mRNA transcript regions. High reproducibility of T-C positional counts was observed (Supplementary Fig. 2a), allowing us to construct high-resolution maps of HDLBP cross-linking profiles. We observed reciprocal binding of HDLBP to CDS and 3′ UTR regions depending on mRNA localization. HDLBP interacted predominantly with the CDS of membrane-bound mRNAs, whereas for cytosolic mRNAs the binding was more prominent in the 3′ UTR (Fig. 2a). To address the relationship between CDS, 3′ UTR binding, and mRNA localization, we calculated the ratio between length-normalized T-C transitions detected in CDS versus 3′ UTR and found that membrane-localized mRNAs had higher ratios compared to cytosolic transcripts (Fig. 2b). In addition, the contribution to the total T-C crosslink signal mostly originated from high CDS binding for membrane-bound mRNAs (Fig. 2c). Comparison between T-C transition patterns in membrane-bound mRNAs IGF2R and

COL2A1, and cytosolic HNRNPUL1 and FTL transcripts (Fig. 2d and Supplementary Fig. 2b), supported the conclusions from the transcriptome-wide analysis. In summary, HDLBP target mRNAs are primarily membrane-localized and bound in the CDS, while in cytosolic mRNAs HDLBP binding is on average equal between CDS and 3′ UTR.

**RNA-recognition elements for HDLBP binding are more common in membrane-bound mRNAs**. We next explored the primary sequence characteristics underlining the specificity of HDLBP for membrane-bound mRNAs. We ranked crosslinked seven-mers by their median crosslink signal per transcript (Supplementary Fig. 3a) and by the frequency of T-C crosslink positions relative to all crosslinked seven-mers (Fig. 3a). These two metrics were compared between CDS and 3′ UTR of mRNAs either localized to the membrane or the cytosol. We found that HDLBP is most frequently crosslinked to CU-containing seven-mers with a variable number of UUC/UC/CUU/CU repeats, located in the CDS of membrane-bound mRNAs (Fig. 3a, b and Supplementary Fig. 3a).

**Fig. 1 HDLBP binds to ER-targeted mRNAs. a** Quantification of membrane and cytosolic mRNA localization in HEK293 cells by sequencing. Histogram of log2-transformed membrane enrichment (ratio between the membrane and cytosolic read counts, mean of two replicates). Cutoffs of 0.5 and 1.5 (gray lines) were chosen to classify membrane-bound ($n = 1157$), cytosolic ($n = 6139$) and mRNAs with undefined localization ($n = 193$). mRNAs with a TPM of at least ten were included in this analysis. **b** Quality control of mRNA-seq-derived localization. Log2-transformed membrane enrichment was compared between groups of mRNAs classified according to presence of encoded targeting signals (TS): cytosol-localized mRNAs with no targeting signals ($n = 5142$), mRNAs encoding mitochondrial proteins excluding mtDNA-encoded mRNAs ($n = 739$), mRNAs coding for post-translationally targeted tail-anchored transmembrane proteins ($n = 102$), mitochondrial DNA expressed mRNAs encoding mitochondrial proteins ($n = 13$), membrane-localized mRNAs with no known TS ($n = 107$), mRNAs encoding signal peptide (SignalP) containing proteins ($n = 214$) or transmembrane helices ($n = 724$) or both ($n = 282$). mRNAs with a TPM of at least ten were included in this analysis. The lower and upper hinges of box plots correspond to the 25th and 75th percentiles, respectively. Upper and lower whiskers extend from the hinge to the largest or smallest value no further than the 1.5× interquartile range from the hinge, respectively. Center lines of box plots depict the median values. **c** Distribution of HDLBP crosslinks (T-C transitions) throughout the transcriptome as detected by PAR-CLIP in HEK293 cells. **d** Scatter plot of normalized HDLBP cross-linking signal in CDS versus mRNA expression level in non-fractionated cells. mRNAs were classified according to their membrane enrichment in (**a**). Inset above shows density distributions of HDLBP cross-linking signal. **e** HDLBP-bound and unbound mRNAs with respect to their localization. Absolute numbers and percentages of HDLBP-bound mRNAs of total HEK293 cytosolic and membrane-bound mRNAs are shown. **f** HDLBP PAR-CLIP binding enrichment (mean of two biological replicates) for different mRNA classes; cytosol-localized mRNAs with no TS ($n = 1578$), mRNAs encoding mitochondrial proteins excluding mtDNA-expressed mRNAs ($n = 251$), mRNAs coding for post-translationally targeted tail-anchored transmembrane proteins ($n = 30$), membrane-localized mRNAs with no known TS ($n = 74$), mRNAs encoding signal peptide (SignalP) containing proteins ($n = 138$) or transmembrane helices ($n = 492$) or both ($n = 219$). The lower and upper hinges of box plots correspond to the 25th and 75th percentiles, respectively. Upper and lower whiskers extend from the hinge to the largest or smallest value no further than the 1.5× interquartile range from the hinge, respectively. Center lines of box plots depict the median values. **a–f** Source data are provided as a Source Data file.

We asked if HDLBP binding is determined by the differential sequence composition of cytosolic and membrane-bound mRNAs. For this purpose, we determined the frequency of all possible seven-mers within whole 3′ UTR and CDS sequences and compared it to the frequency of HDLBP crosslinked seven-mers. While the top HDLBP crosslinked seven-mers were not among the most frequent seven-mers in the transcriptome, they showed a significantly higher occurrence ($P = 2.7e-07$) in the CDS of membrane-bound compared to cytosolic mRNAs (Supplementary Fig. 3b).

Since HDLBP is composed of 15 KH domains[26,33], we next addressed the possibility that it could recognize longer RREs[40]. Therefore, the difference in the frequency between membrane-bound and cytosolic mRNAs was calculated for the 40 most highly crosslinked k-mers with a length between 4 and 12 nucleotides. Crosslinked k-mers were generally more frequent in sequences of membrane-bound mRNAs than in cytosolic mRNAs (Fig. 3c). In addition, the greatest difference between membrane-bound and cytosolic mRNAs was observed for the longest k-mers (10–12 nt) (Fig. 3c). Therefore, membrane-bound mRNAs contained a significantly higher number of longer high-affinity RREs for HDLBP binding than cytosolic mRNAs (Fig. 3d). On the other hand, the occurrence of crosslinked k-mers was comparable between CDS and 3′ UTR of cytosolic mRNAs (Supplementary Fig. 3c, d), and corresponded to the equal distribution of HDLBP crosslinks between these two transcript regions (Fig. 2c). In summary, the specific binding of HDLBP to the CDS of membrane-bound mRNAs can at least in part be explained by the differential k-mer composition of membrane-bound and cytosolic mRNAs.

**High-affinity multivalent HDLBP interactions are more frequently formed in membrane-bound mRNAs**. Due to the high number of KH domains, we next reasoned that HDLBP could recognize RREs interspaced with unbound nucleotides, giving rise to multivalent interactions. By counting the frequency of most frequently crosslinked four-mers in 40 nt regions around all detected crosslink positions (Supplementary Fig. 3f), we found that the regions with the highest four-mer frequency also showed the highest crosslink values within these regions (Fig. 3e). Therefore, the highly multivalent HDLBP binding sites also

resulted in high-affinity interactions. In addition, the average positional crosslink signal within regions with the highest multivalency revealed that several positions upstream ($-13$ nt) and downstream ($+16$ nt and $+20$ nt) of the crosslink were specifically crosslinked (Fig. 3f). Therefore, high-affinity HDLBP sites contain 3–4 RNA elements positioned several nucleotides apart giving rise to binding sites of ~40 nt in length. Notably, multivalent interactions resulted in high-affinity interactions both in the CDS and 3′ UTR of membrane-bound and cytosolic mRNAs, respectively (Supplementary Fig. 3e).

To explore the potential of membrane-bound and cytosolic mRNAs to form multivalent interactions with HDLBP, we next calculated the occurrence of four-mers within 30-nt sliding windows of mRNA sequences. We observed that the local frequencies of HDLBP crosslinked four-mers were significantly higher in CDS of membrane-bound than in cytosolic mRNAs (Fig. 3g). This effect was absent for a group of unbound four-mers, confirming that membrane-bound CDS contain a high density of local HDLBP recognition elements that give rise to multivalent interactions.

To determine the RNA substrate specificity in more detail, we expressed and purified recombinant full-length HDLBP (FL) and protein variants (A–D) harboring different sets of KH domains (Fig. 3h and Supplementary Fig. 3i). Using a fluorescent anisotropy assay, we determined the apparent dissociation constants (Kd) for the respective protein–RNA substrate combinations, which are summarized in Fig. 3h. The full-length HDLBP bound with high-affinity to regions of TFRC mRNA (TFRC_1 and _2), which were selected based on our PAR-CLIP data. Mutations in one of the TFRC sites (TFRC_1_mut) reduced binding. Using artificial sequences, we could show that HDLBP interacted strongly with CCU- and CUU-oligomers, whereas affinity to a CAA oligomer was about tenfold reduced. In addition, HDLBP, albeit with reduced affinity, is bound to a region in the ES6 region of the 18 S rRNA. Interestingly, HDLBP construct B, comprising KH domains 5 through 9, bound with nearly similar affinities to the tested mRNA and 18S rRNA sequences as the full-length protein, suggesting that the central part of HDLBP is important for recognition and interaction with different RNA substrates. The 5 C-terminal KH domains HDLBP showed reduced binding to the TFRC mRNA region when compared to full-length HDLBP. For the N-terminal HDLBP

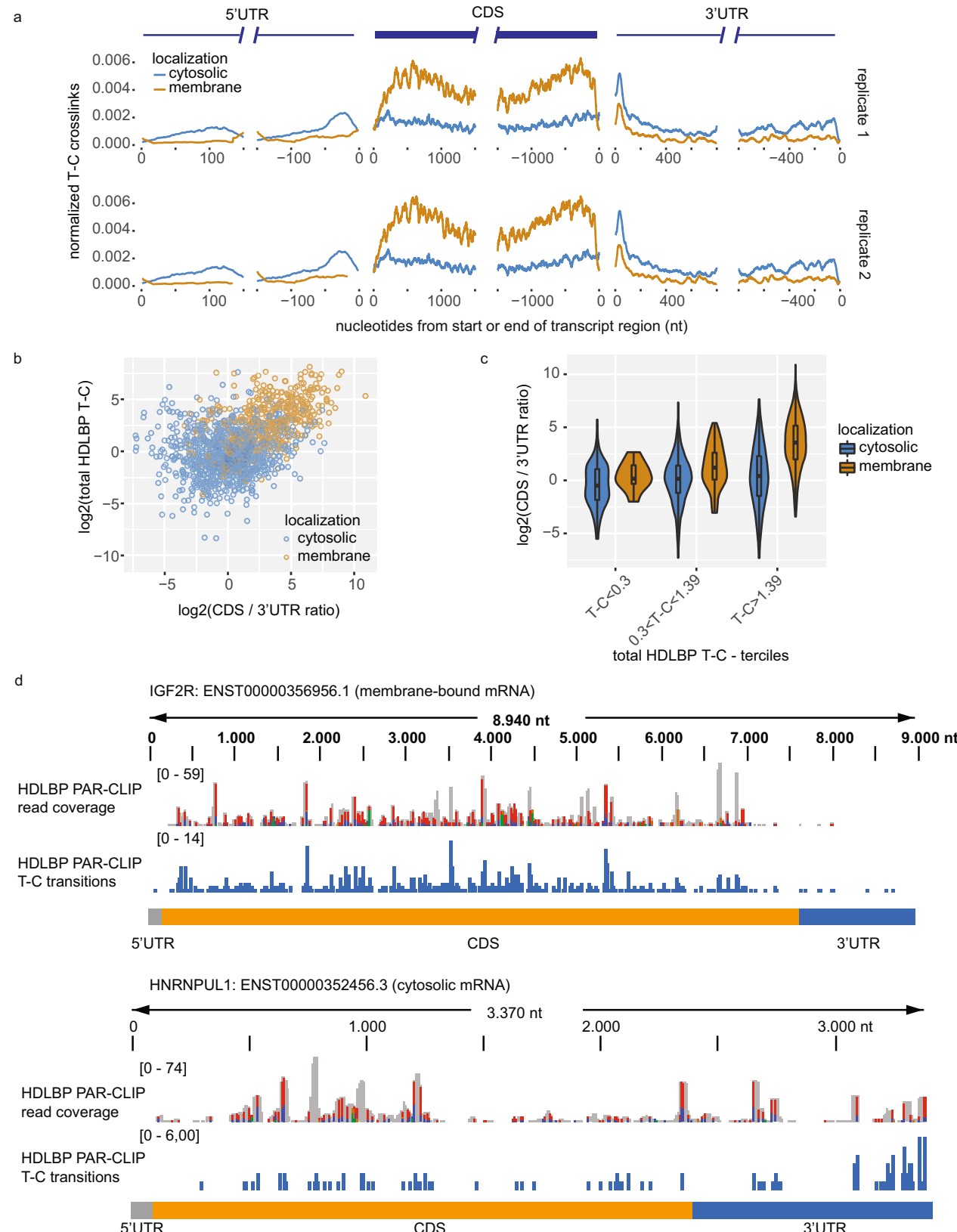

fragment A (KH0–KH4) we were unable to identify a high-affinity RNA substrate.

To understand the impact of multivalent interactions on HDLBP binding affinity, we next performed the in vitro HDLBP binding assay using RNA molecules with different numbers of bound four-mer motifs within a longer (34 nt) sequence (Fig. 3i).

We determined significantly lower Kd values for synthetic RNA molecules containing three HDLBP-bound four-mers (CUUC or UCUU) than those that harbored only 2. Therefore, HDLBP (KH5-9) has ~3–5-fold higher affinity for RNAs with higher multivalent potential. These findings strongly support our conclusions from PAR-CLIP analysis and confirm that HDLBP

**Fig. 2 Differential HDLBP binding to ER-targeted and cytosolic transcripts. a** Meta-transcript analysis of HDLBP cross-linking signal for cytosolic and membrane-bound mRNAs. T-C transition signal per nucleotide was normalized for library size (T-C per million). Transcripts with at least 5 T-C per million were included in this analysis. T-C per million was scaled to the maximum T-C signal per transcript. At each position, the mean scaled T-C signal was plotted. **b** Scatter plot of the ratio of HDLBP crosslinks in the CDS vs. 3′ UTR plotted against total crosslinks per mRNA for cytosolic and membrane-bound mRNAs. **c** mRNAs were split into three groups according to total HDLBP crosslinks per mRNA ($n = 16$: TC <0.3, membrane, $n = 65$: 0.3 < TC <1.39, membrane, $n = 35$:, TC >1.39, membrane, $n = 279$: TC <0.3, cytosolic, $n = 379$: 0.3 < TC <1.39, cytosolic, $n = 144$: TC >1.39, cytosolic) and the distributions of the CDS vs. 3′ UTR ratio of HDLBP crosslinks were plotted. The lower and upper hinges of box plots correspond to the 25th and 75th percentiles, respectively. Upper and lower whiskers× extend from the hinge to the largest or smallest value no further than the 1.5× interquartile range from the hinge, respectively. Center lines of box plots depict the median values. **d** Browser representation of PAR-CLIP read coverage and T-C transitions in IGF2R (membrane-bound) and HNRNPUL1 (cytosolic) mRNAs. Reads were mapped to human mRNA sequences. 5′ UTR, CDS, and 3′ UTR regions are indicated. Transcript IDs are indicated. **a–d** Source data are provided as a Source Data file.

binds with high affinity to long RNA regions with multivalent interactions.

**HDLBP interacts with the translational apparatus**. Since HDLBP showed specific interactions with the CDS of membrane-bound mRNAs, we next explored the possibility that it plays a role in the translational regulation of ER-targeted mRNAs. HDLBP PAR-CLIP showed binding to 18S rRNA with two major binding sites located in expansion segment 6SB and helix 16 (Fig. 4a). These sites likely play a regulatory role in translation initiation and elongation[41,42] due to their proximity to the binding sites of eukaryotic initiation and elongation factors (EIF4A, EIF4G1, and EEF2) (Supplementary Fig. 4a). Next, we examined HDLBP binding to the 7SL RNA component of the signal recognition particle (SRP), an RNP required for co-translational targeting of nascent polypeptide-associated complexes to the ER membrane. In 7SL RNA, we observed HDLBP contacts that were distinct from the binding patterns of other RBPs (IRE1, SSB, and MOV10) (Fig. S4B). HDLBP crosslinks were located in helices (5d–f, 6, 8a) of the large (S) domain and at the unpaired uridines of the small Alu region (Supplementary Fig. 4c). The percentage of SRP over total T-C crosslinks was comparable to other known or expected 7SL RNA interactors (IRE1 and SSB) and ~10-fold higher than for MOV10, which does not strongly bind 7SL RNA (Supplementary Fig. 4d). In summary, the HDLBP interacts with the 40S subunit of the ribosome at positions that are also close to the SRP contact sites (Fig. 4b).

In order to ensure that these contacts resulted from stable interactions, we confirmed various HDLBP RNA targets by RNA immunoprecipitation (RIP) experiments (Fig. 4c). Results corresponded to the conclusions of PAR-CLIP experiments with 7SL RNA showing a moderate enrichment, while several membrane-bound mRNAs (YWHAZ, ATP1A1, CD46, and IGF2R) showed high enrichment, whereas an mtDNA-encoded mRNA (MT-CO1) showed no enrichment. These results supported the conclusion that HDLBP forms stable interactions with RNA species identified by PAR-CLIP.

To validate the interaction with the translational apparatus by an orthogonal method, we next profiled HDLBP-proximal proteins using BioID[43], a proximity labeling assay in HEK293 cells expressing HDLBP fused to a promiscuous biotin ligase BirA* (Supplementary Fig. 1d). We confirmed by RIP that BirA*-FLAG-HDLBP was also bound to several transcripts that we previously detected to be bound by the FLAG/HA-tagged HDLBP (Supplementary Fig. 4f). Reproducible identification of HDLBP-proximal proteins (Supplementary Fig. 4e, g and Supplementary Data 4) overlapped ~50 % with the previously published BioID experiment (Supplementary Fig. 4h)[44] (Fig. 4d, e). We found that the top enriched proteins were involved in translation and included translation initiation factors (EIF4G1, EIF4B, EIF5, and EIF4E2) (Fig. 4d, e and Supplementary Fig. 4i),

chaperones and chaperonins (HSPA1A, HSPA8, and CCT8), SRP components (SRP68) and ribosomal proteins (RPS3A and RPS10) of the small subunit (SSU). We validated the potential SSU interaction by orthogonal anti-FLAG co-immunoprecipitation experiments and found that the co-immunoprecipitates contained the SSU component of the RPS6 but very little amount of RPL7 (Fig. 4f). This finding agrees with specific 18S rRNA crosslinks detected by PAR-CLIP and HDLBP binding to an 18S rRNA sequence as shown in Fig. 3h. Taken together, these results showed that HDLBP interacts with the translational apparatus including the SRP and is localized in the proximity of ribosome-associated factors.

**HDLBP promotes translation at the ER membrane**. Based on previous findings[45–49] and the specific interaction of HDLBP with the CDS of membrane-bound mRNAs and the ribosome observed in our study, we next addressed the function of HDLBP in ER-associated translation. We thus measured the process of active translation in the presence and absence of HDLBP by generating two CRISPR/Cas9 HDLBP knockout (KO) cell lines (Fig. 5a). The HEK293 HDLBP knockout cells showed no apparent growth defect and electron microscopy imaging of the ER revealed no morphology changes (data not shown). To quantify translation efficiency in the KO and wild-type (WT) conditions we generated ribosome profiling datasets, which showed high read periodicity and dominant in-frame P-site coverage in the CDS (Supplementary Fig. 5a, b). Since this dataset was obtained from non-fractionated cells, we asked if our experiment sufficiently captured the ribosome footprints on ER-bound mRNAs. As expected we observed low footprint density for SP and TM-containing mRNAs in the region downstream of the start codon, until the emergence of both targeting signals from the ribosome (Supplementary Fig. 5c), confirming that the data recovered ER-bound ribosome complexes with near-nucleotide resolution.

To address the differences in translation efficiency upon HDLBP KO we compared groups of membrane-bound mRNAs (Supplementary Data 5), which were classified according to the number of HDLBP crosslinks in the CDS. The highest decrease in translation efficiency was observed for mRNAs that had the largest number of HDLBP crosslinks in the CDS, suggesting that HDLBP interactions with membrane-bound mRNAs promoted translation (Fig. 5b) but did not affect ER mRNA localization (Supplementary Fig. 5d). We validated these findings by quantifying protein synthesis using pulsed stable isotope labeling with amino acids in cell culture (pSILAC)[50] in combination with subcellular fractionation (Supplementary Data 6). The absence of HDLBP generally resulted in a decrease in protein synthesis of proteins encoded by membrane-bound mRNAs (Fig. 5c) and the extent of decrease depended on the level of HDLBP cross-linking signal (Fig. 5d and Supplementary Fig. 5e). Therefore, HDLBP is required for efficient protein synthesis of its target mRNAs. Since

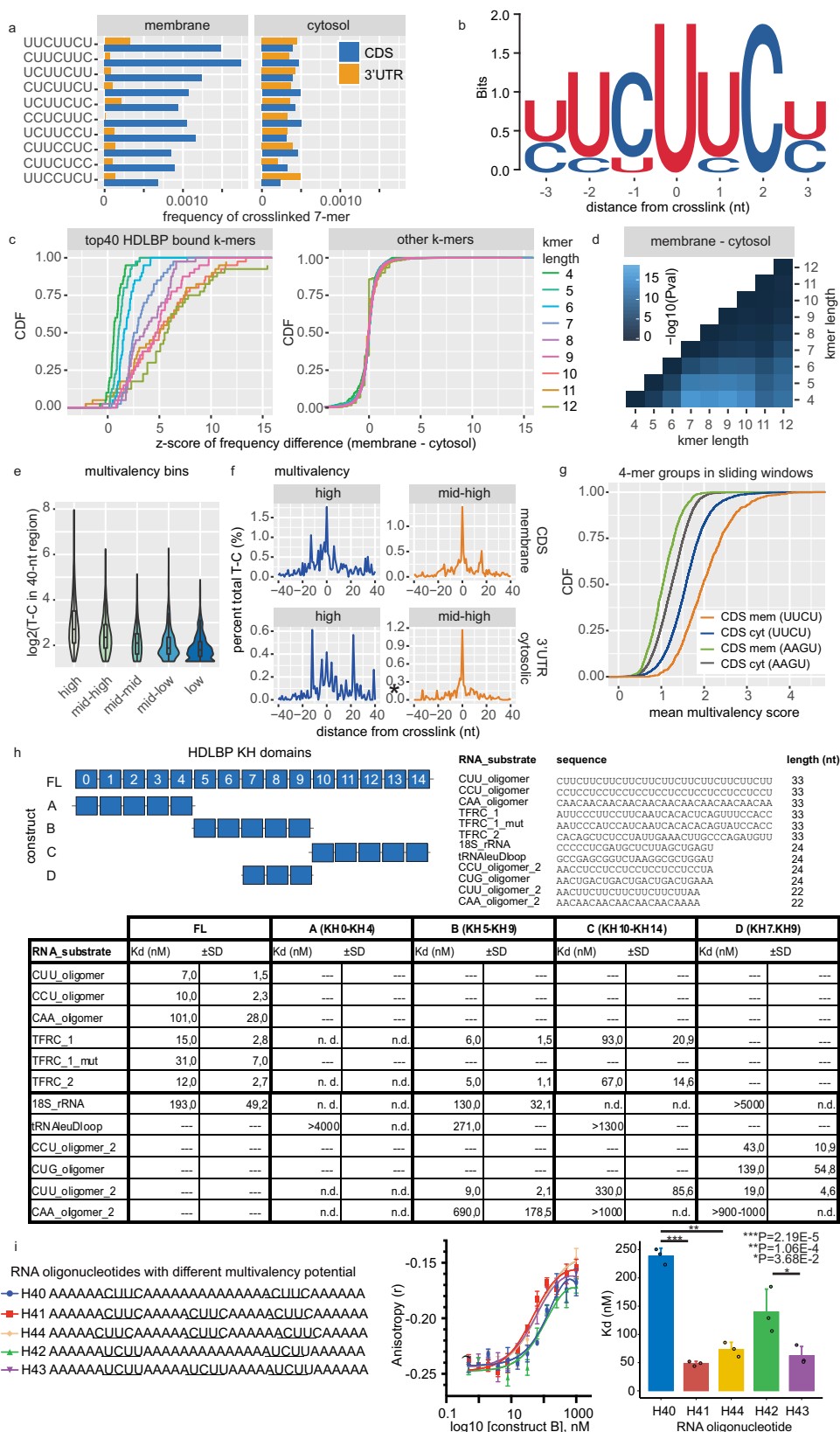

the ER membrane is the primary site of translation of secretory proteins, we next asked if HDLBP influences their secretion. For this purpose, we expressed the secreted *Gaussia* luciferase (Gluc) and alkaline phosphatase (SEAP) in HEK293 cells in WT and KO conditions and quantified enzyme activity in the culture medium. Gluc and SEAP activity was significantly decreased upon HDLBP

KO by 20–40%, showing that HDLBP depletion reduces the secretion of the two reporter proteins (Fig. 5e, f and Supplementary Fig. 5f).

Since the depletion of HDLBP reduced secretion, we next tested the impact of HDLBP overexpression. To this end we transiently transfected FLAG/HA-HDBLP into HEK293 and

**Fig. 3 HDLBP specificity for membrane-bound mRNAs. a** Frequency of top ten HDLBP crosslinked seven-mers located either in 3′ UTR or CDS of membrane-bound and cytosolic mRNAs. **b** Sequence logo of top five HDLBP crosslinked seven-mers ranked according to their frequency among all detected crosslinked seven-mers. **c** Distribution of $z$-scores calculated from differences in the frequency of all possible k-mers within membrane-bound and cytosolic CDS and 3′ UTR sequences. For each k-mer length, this analysis was performed for the group of top 40 HDLBP crosslinked k-mers (left) and all other k-mers (right). **d** $p$ values of pairwise Wilcoxon rank-sum test between $z$-scores obtained for top 40 bound HDLBP k-mers as described in (**c**). **e** HDLBP multivalency analysis in $+40/-40$ nt regions around cross-linking sites. To evaluate the T-C binding affinity as a function of multivalency of HDLBP binding sites, we binned the multivalency scores for the top ten enriched four-mers within the 40-nt regions into five categories (group sized from highest to lowest score groups, $n = 994$, $n = 973$, $n = 673$, $n = 1036$, $n = 1308$). The total normalized T-C transition signal over the $+40/-40$ nt regions was then plotted for all five categories. Lower and upper hinges of box plots correspond to the 25th and 75th percentiles, respectively. Upper and lower whiskers extend from the hinge to the largest or smallest value no further than the 1.5× interquartile range from the hinge, respectively. Center lines of box plots depict the median values. **f** Analysis of the percentage of total T-C transitions for every nucleotide position within the $+40/-40$ nt region for each multivalency bin. The two bins with the highest multivalency scores are shown and correspond to (**e**). **g** Comparison of mean multivalency scores between differentially localized mRNAs in their CDS. A positive set (a four-mer group consisting of top ten HDLBP crosslinked four-mers, UUCU) and a negative set (AAGU) with no HDLBP enrichment. Occurrence of these four-mer groups were counted in 30-nt sliding windows and the mean score per transcript was computed. Mean distribution was then compared between different localized CDS by Wilcoxon rank-sum tests. **h** Apparent dissociation constants of recombinant GST-HDLBP fragments (constructs A though D) and full-length protein (FL), schematically shown, for different RNA oligonucleotides as determined by fluorescence anisotropy binding assays. Dissociation constants that were measured but could not be determined are indicated with "n.d." and interaction that were not measured by "—". **i** Fluorescence anisotropy binding assays for RNA oligonucleotides with different number of HDLBP binding four-mers (left, H40-44). GST-HDLBP construct B (KH5-9) was incubated with FAM-labeled RNA oligonucleotides, anisotropy measured and $K_D$ determined from the binding curves (middle panel). For each oligonucleotide, three independent $K_d$ values were determined. Significant differences in $K_d$ values were evaluated using two-sided $t$-test and are indicated with asterisks (*$P < 0.01$, *$P < 0.05$). Data were presented as mean values ± SD. **a–i** Source data are provided as a Source Data file.

HEK293 HDLBP KO cells. We also stably overexpressed HDLBP in A549 cells using a piggybac transposon carrying HDLBP (Fig. 5g).

To test the effect on secretion, we transfected the SEAP reporter construct into HEK293 and A549 cells with differing HDLBP expression levels (Fig. 5h). As expected, KO of HDLBP in HEK293 and A549 cells reduced the secretion of SEAP, whereas overexpression of HDLBP in HDLBP knockout cells rescued SEAP secretion to the levels observed in wild-type cells. Interestingly, the re-introduction of HDLBP into HEK293 and A459 cells resulted in increased secretion of about 1.15- and 2-fold, respectively (Fig. 5h) confirming that HDLBP expression levels directly influence the extent of SEAP secretion.

We next asked if the absence of HDLBP affects ribosome occupancy on membrane-bound mRNAs. For this purpose, we displayed the footprint coverage around known targeting signals (Fig. 5i and Supplementary Fig. 5g) and found that the absence of HDLBP resulted in lower ribosome density immediately downstream of the region encoding SPs or the first TM domain. This suggests that HDLBP contributes to ribosome elongation arrest, which is required for co-translational targeting and efficient translocation of the nascent peptide.

**HDLBP crosslinks to CU/UU-containing codons in mRNAs and tRNAs decoding these codons**. We next interrogated our ribosome profiling datasets to quantify ribosome occupancy per codon in WT and KO conditions. While we found that upon HDLBP KO several codons (including UUC/Phe, UUU/Phe, CUC/Leu, and CUU/Leu) were on average slightly more occupied by the ribosome and the P-site and E-site, this increase was not statistically significant. Therefore, global ribosome stalling in the KO condition may be measurable but could be an indirect effect of HDLBP depletion. Analysis of normalized HDLBP PAR-CLIP signal in codons (Supplementary Fig. 6a), identified the same codons (UUC/Phe, CUC/Leu, and CUU/Leu) to be among the most highly bound.

In addition, we analyzed HDLBP binding to tRNAs by PAR-CLIP enrichment as well T-C transition specificity for each tRNA cross-linking position (Supplementary Data 7) and observed that four leucine isotype tRNAs were among the top 15 enriched tRNAs in the HDLBP-bound pool (Fig. 6b). Their cross-linking

sites were located in variable and D loops (Fig. 6c and Supplementary Fig. 6b) and corresponded to HDLBP mRNA binding motifs (UCUUC). Interestingly, the codons decoded by these tRNAs tend to be more occupied in HDLBP-depleted cells (Fig. 6a), suggesting that HDLBP tRNA binding enables more efficient tRNA decoding likely by facilitating tRNA recycling or reduction of tRNA diffusion from the ribosome, as proposed previously[34].

**Absence of HDLBP results in lower proliferation and tumor formation capacity**. HDLBP function in ER translation and secretion could impact the production of mitogens, growth factors, receptors and extracellular matrix, and consequently greatly influence cell proliferation, differentiation, migration, and invasion. To test this, we established a CRISPR/Cas9-induced HDLBP KO in the lung adenocarcinoma (LUAD) derived cell line A549. In 2D cultured A549 cells, adhesive growth was markedly (by 40–50%) reduced by HDLBP deletion (Fig. 7a). Likewise, HDLBP KO interfered with the 2D migration of A549 cells, as indicated by severely reduced wound closure in scratch migration analyses (Fig. 7b). The surface expression of the transmembrane glycoprotein CD71 encoded by the HDLBP-bound TFRC mRNA (Fig. 3h) was reduced upon HDLBP KO (Fig. 7c and Supplementary Fig. 7a, b). In contrast, overexpression of HDLBP in A549 cells resulted in accelerated wound closure (Fig. 7d). Collectively, these findings suggested that HDLBP substantially influences the oncogenic properties of tumor cells. To test this in vivo, parental and HDLBP KO A549 cells were stably transduced with iRFP (infrared fluorescent protein) and viable cells were subcutaneously (s.c.) injected into athymic (FOXN1$^{nu/nu}$) nude mice to monitor tumor initiation and growth (Fig. 7e). Homogenous s.c. application of both cell populations was validated by non-invasive near-infrared imaging of iRFP immediately post-injection (Supplementary Fig. 7c, 0 days post-injection). Strikingly, tumor initiation and growth were severely reduced by HDLBP deletion (Fig. 7e and Supplementary Fig. 7c). In contrast to parental cells, which formed tumors in all eight analyzed animals, HDLBP KO cells formed palpable tumors in only three of eight animals (Supplementary Fig. 7c). In accordance, tumor volume and final tumor mass were significantly reduced by

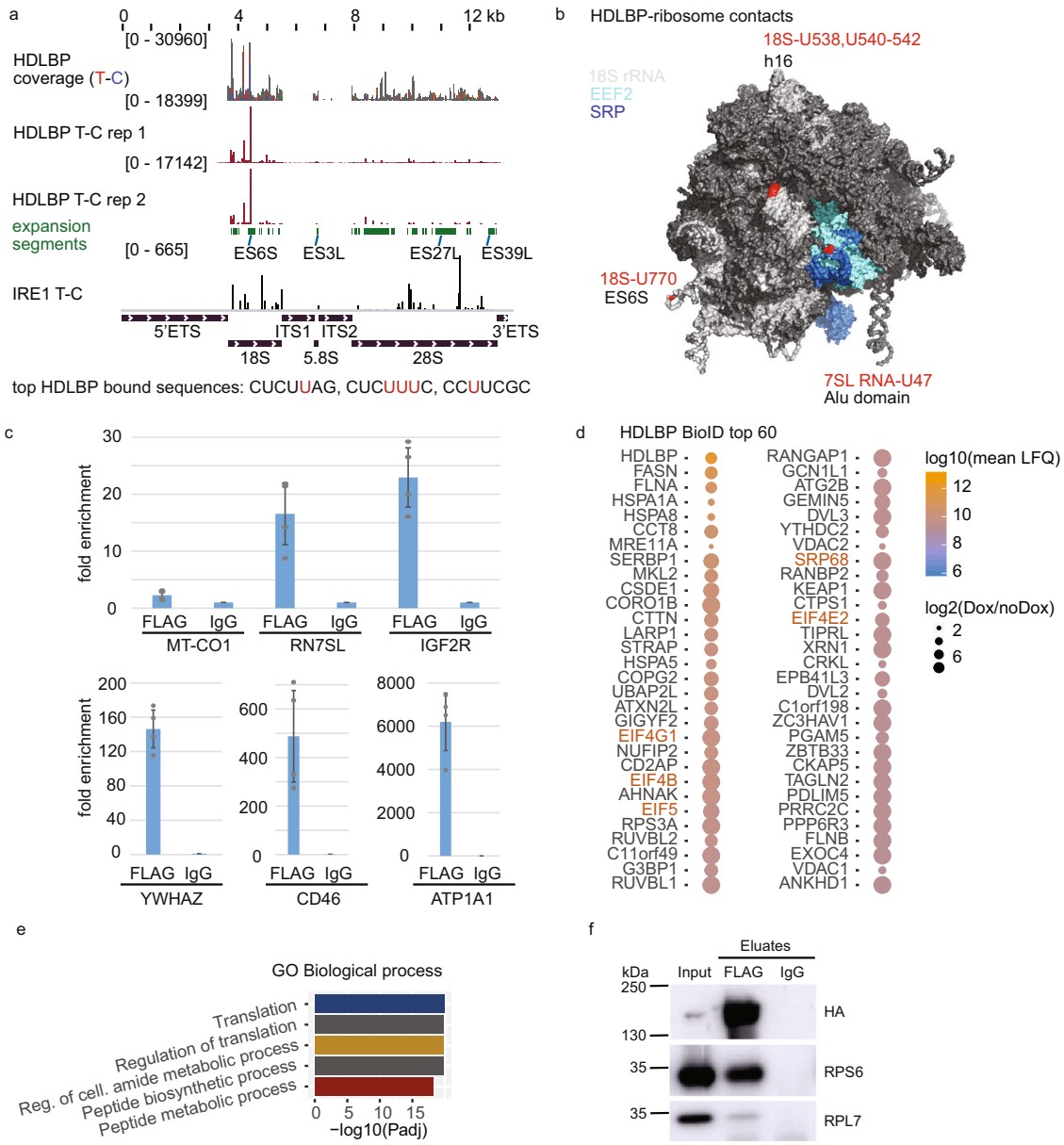

**Fig. 4 HDLBP interacts with the translational apparatus. a** PAR-CLIP coverage and crosslinks detected in pre-rRNA regions. For comparison, results for IRE1 PAR-CLIP are included. Expansion segment positions are indicated in green. **b** Structures of the human 80S ribosome (PDB: 4V6X) and the SRP-ribosome complex (PDB: 3JAJ) were juxtaposed and HDLBP rRNA and 7SL RNA crosslinked nucleotides were mapped (indicated in red). **c** RNA immunoprecipitation was performed with FLAG/HA-HDLBP as bait. Co-precipitating RNAs were detected by qRT-PCR. Average fold enrichment (anti-FLAG vs. IgG control) from four replicates was calculated with error bars indicating standard deviation. Results are shown for 7SL RNA, IGF2R, YWHAZ, CD46, and ATP1A1 are shown, along with the mtDNA-encoded mRNA (MT-CO1) as a negative control. **d** BioID analysis of proteins in proximity to BirA-FLAG-HDLBP. The top 60 enriched proteins (LFQ(Dox) vs. LFQ(noDox) >3) were ranked according to mean LFQ (three replicates of Dox samples). The size of the dot corresponds to the enrichment value. **e** Gene Ontology enrichment analysis of 249 enriched BioID proteins. Adjusted *p* values for the top five enriched categories are shown. **f** FLAG/HA-HDLBP was co-immunoprecipitated with either anti-FLAG or IgG antibodies. Western analysis of input lysates (0.25%) and eluates (19%) was performed with antibodies as indicated. **a–f** Source data are provided as a Source Data file.

HDLBP deletion (Fig. 7f), indicating that HDLBP is essentially involved in tumor initiation and growth.

To determine how HDLBP KO influenced gene expression, RNA abundance was monitored by RNA sequencing in final tumors (Supplementary Fig. 7d). In the absence of HDLBP, a high number of mRNAs was found to be up ($n = 1039$) and downregulated ($n = 700$), suggesting that HDLBP has a great impact on gene expression, presumably due to substantially impaired cell signaling and tumor-stroma cross-talk (Supplementary Fig. 7e and Supplementary Data 8). Strikingly, the protein products of mRNAs decreased by HDLBP KO were

highly enriched for secreted proteins (e.g., collagens and matrix metalloproteases, MMP2) that function in biological processes such as "extracellular matrix organization", "cell adhesion", and "tissue development" (Fig. 7g). These findings supported the view, that HDLBP influences protein output, expression, and/or turnover of transcripts encoding factors essentially involved in modulating the composition of and cross-talk within the tumor-stroma landscape. To evaluate the direct control of deregulated mRNAs by HDLBP, we compared the fold changes between WT and KO xenograft tumors for subgroups of mRNAs classified according to their PAR-CLIP signal in

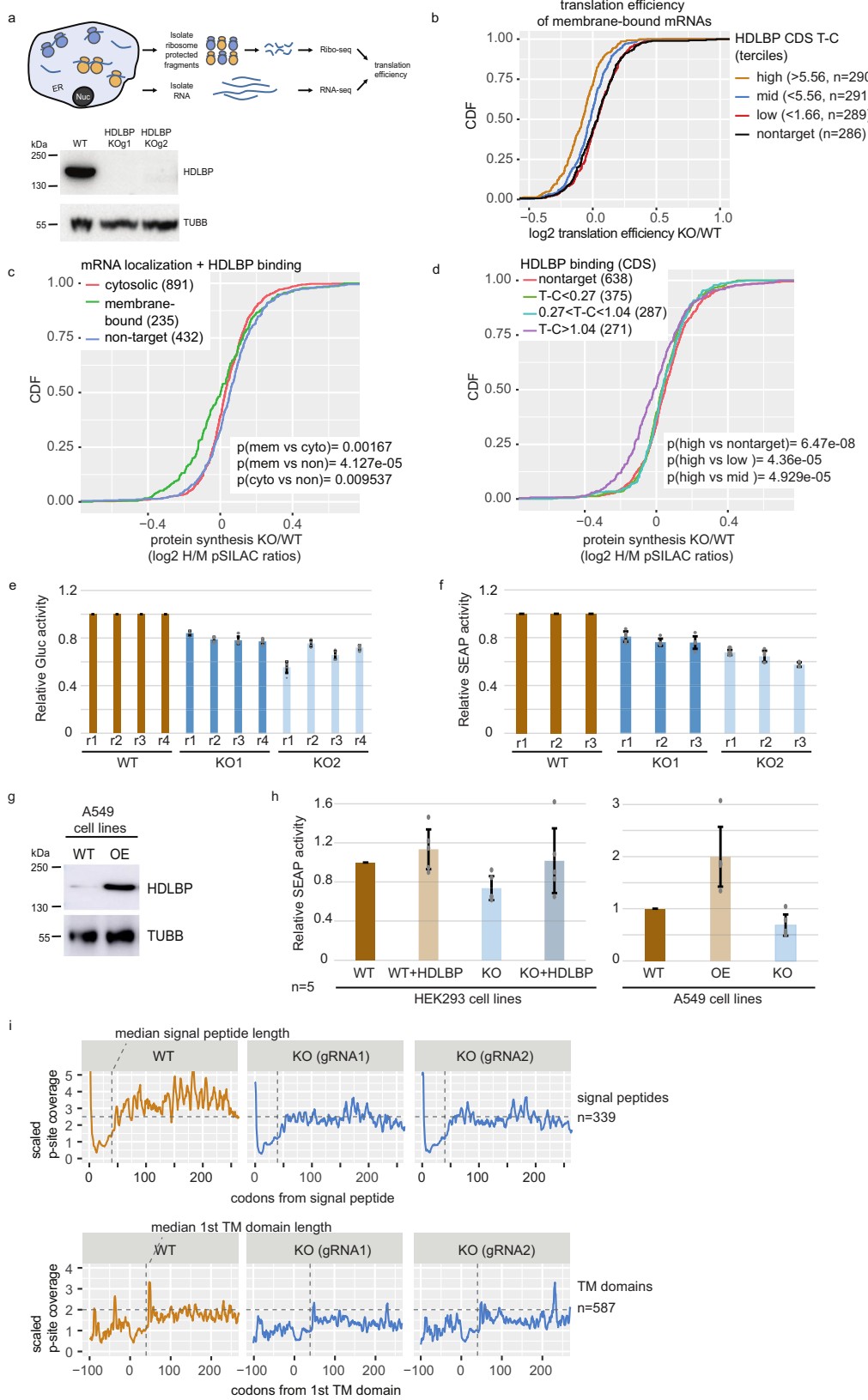

HEK293 cells. We found that changes in mRNA expression of HDLBP target mRNAs (Supplementary Fig. 7f) and membrane-bound mRNAs (Fig. 7h) were significantly decreased in xenografts lacking HDLBP than for non-targets or cytosolic mRNAs. Although it remains to be addressed if HDLBP influences mRNA turnover by modulating ER-associated translation, these findings suggest that next to indirect regulation of gene expression, HDLBP also directly influences mRNA abundance, potentially by modulating ER-associated mRNA translation. In sum, the presented findings provide strong evidence that HDLBP is an important modulator of tumor progression influencing the expression of secreted

**Fig. 5 HDLBP promotes ER translation and synthesis of secretory and transmembrane proteins. a** Schematic overview of the ribosome profiling experiment in HEK293 parental and HDLBP KO cells. Western analysis shows the absence of HDLBP in KO cells. **b** Membrane-bound HDLBP target mRNAs were split into three similarly sized groups based on their cross-linking signal in the CDS (indicated in parentheses). Differences in translation efficiency (HDLBP KO vs. WT) were compared between groups. A two-sided Wilcoxon rank-sum test was used to test for significance. **c** pSILAC analysis of newly synthesized proteins in HEK293 parental and HDLBP KO cells. SILAC heavy vs. medium ratios (H/M) reflect changes in protein synthesis upon HDLBP KO and were quantified in membrane fractions. Proteins were split into three similarly sized groups based on their cross-linking signal in the CDS (indicated in parentheses). SILAC ratios were compared between groups using a two-sided Wilcoxon rank-sum test. **d** Proteins were split into three similarly sized groups based on their PAR-CLIP signal and membrane localization (indicated in parentheses). SILAC ratios were compared between groups using a two-sided Wilcoxon rank-sum test. **e** Parental and HDLBP KO cells were transfected with secreted *Gaussia* luciferase (Gluc) construct. Gluc activity was quantified in the medium and normalized to the intracellular Firefly luciferase (Fluc) activity. Each of the four replicate experiments (r1–r4) was carried out with five technical replicates. Data were presented as mean values ± SD. **f** Parental and HDLBP KO cells were transfected secreted alkaline phosphatase (SEAP) construct. SEAP signal was quantified in medium and normalized to the intracellular Firefly luciferase (Fluc) activity. Each experiment was carried out in five technical replicates. Data were presented as mean values ± SD. **g** Western analysis of A549 cells stably transfected with construct overexpressing HDLBP (OE). **h** SEAP activity, relative to Fluc, was measured in HEK293 and A549 cells with different HDLBP protein levels (KO knockout, WT wild-type, HDLBP/OE overexpression). The experiment was carried out five times with at least five technical replicates. Data were presented as mean values ± SD. **i** Ribosome P-site coverage around targeting signals (signal peptides and transmembrane helices) was compared between HEK293 HDLBP knockout (KO) vs. parental (WT) cells. P-site coverage was scaled to the coverage in codons 20–40 of each mRNA. A rolling mean of 5 nt was used to smooth the profiles. Absolute numbers of analyzed mRNAs are given, median signal peptide and first transmembrane helix lengths are indicated with a vertical dotted line. **a–i** Source data are provided as a Source Data file.

## Discussion

In this work, we characterized the function of HDLBP in the context of translation at the ER. While ~80% of membrane-bound mRNAs were found to be high-affinity substrates of HDLBP (Fig. 1e), they also showed predominant binding in the CDS (Figs. 1c, 2b). In contrast, only ~40% of all cytosolic mRNAs contained HDLBP binding sites but showed significantly lower affinity and were more randomly distributed between CDS and 3′ UTRs (Fig. 2b). Investigation of the primary sequences of ER-targeted and cytosolic mRNAs revealed that the HDLBP-bound CU-containing motifs were more frequent in membrane-bound mRNAs (Fig. 3a), which also showed high codon frequency for hydrophobic amino acids, such as Leu (CUC and CUU), Ile (AUC), Phe (UUC), and Val (GUC), commonly present in signal peptides and transmembrane helices[51].

While the previously reported CHHC/CHYC binding motif[35] matched our findings, we further found evidence that the frequency of longer HDLBP-bound motifs up to 12 nt in length is higher than for shorter motifs in ER-bound than in cytosolic mRNAs (Fig. 3c, d). Therefore, HDLBP may have evolved to specifically recognize membrane-bound mRNAs by making use of their differential sequence composition. Multiple KH domains may allow HDLBP to recognize ER-bound mRNA through its interaction with long heterogeneous RREs and/or with multi-partite motifs resulting in multivalent high-affinity binding regions, as observed for other RBPs[28,52–54]. We thus quantified highly multivalent HDLBP sites and found that they correlated with high binding affinity, were ~40 nt long and contained on average 3–4 UC/CU-containing four-mers positioned several nucleotides apart. This may reflect the binding of 3–4 functional domain modules within the HDLBP protein structure to multiple RREs. Using in vitro binding assays (Fig. 3i), we confirmed these observations and conclude that multivalent interactions indeed result in the higher affinity of HDLBP to long RNA-binding regions. Further studies will provide insight into which domains or their combinations are responsible for the recognition of long multivalent sites and the formation of HDLBP-containing mRNPs. The recently identified C/U-repeat-containing SECReTE motif[17] supports the existence of long functional sequences in yeast mRNA that are required for the secretion of encoded proteins and very likely represents the RNA-binding sites of the HDLBP yeast orthologue Scp160p.

In addition, we report that HDLBP is interacting with ER-associated mRNAs and promoting their translation resulting in increased secretion of the synthesized proteins (Fig. 5), which is in line with previous findings[34,35]. Since we found high-affinity HDLBP binding sites not only in ER-targeted mRNAs but also in membrane-localized mRNAs with no known or predicted targeting signals (Fig. 1d), HDLBP may significantly contribute to the efficient translation of both SRP-dependent and -independent mRNAs[8–12]. Profiling of SRP-dependent and -independent mRNAs in mammalian cells will contribute to this understanding in future studies.

Furthermore, we show that HDLBP absence results in decreased ribosome elongation arrest around targeting signals (Fig. 5g). This process is required for efficient targeting and translocation of nascent peptides to the ER lumen, as well as to prevent misfolding, aggregation, and ER-associated degradation[55]. HDLBP-mediated local ribosome slowdown may thus promote targeting, translocation, and accurate folding of protein domains encoded by HDLBP-bound regions. This possibility is strongly supported by the detected h16 and ES6SB 40S HDLBP interactions, which are in proximity to the EEF2 and EIF4G2 binding sites[41] and could influence ribosome elongation arrest. In addition, low-affinity HDLBP contacts detected in the 7SL RNA region, which is required for the delay in GTP hydrolysis during elongation arrest, and membrane targeting activity[56] also support this notion. Finally, we identified key chaperones and chaperonins in close proximity to HDLBP, supporting its role in protein folding, as previously suggested for its yeast orthologue[36].

Mechanistically, we propose that HDLBP interacts with the translational apparatus by binding to the mRNA and the small ribosomal subunit as well as to other ribosome-associated factors. High-affinity mRNA binding contributes to the formation of elongation-arrested cytosolic RNCs, allowing their efficient localization to the ER membrane, translocon handover, and/or enabling proper folding of the nascent peptide. Since HDLBP is most likely bound to the mRNA downstream from the elongating ribosome, it could be sequestered from the mRNA via ribosome collisions possibly via tRNA and/or ribosome ES6SB-dependent mechanism, which remains to be elucidated. We speculate that HDLBP is bound to the mRNA only during the primary round of translation after which it is removed. This step may allow the mRNA molecule to localize to the ER, where additional rounds of

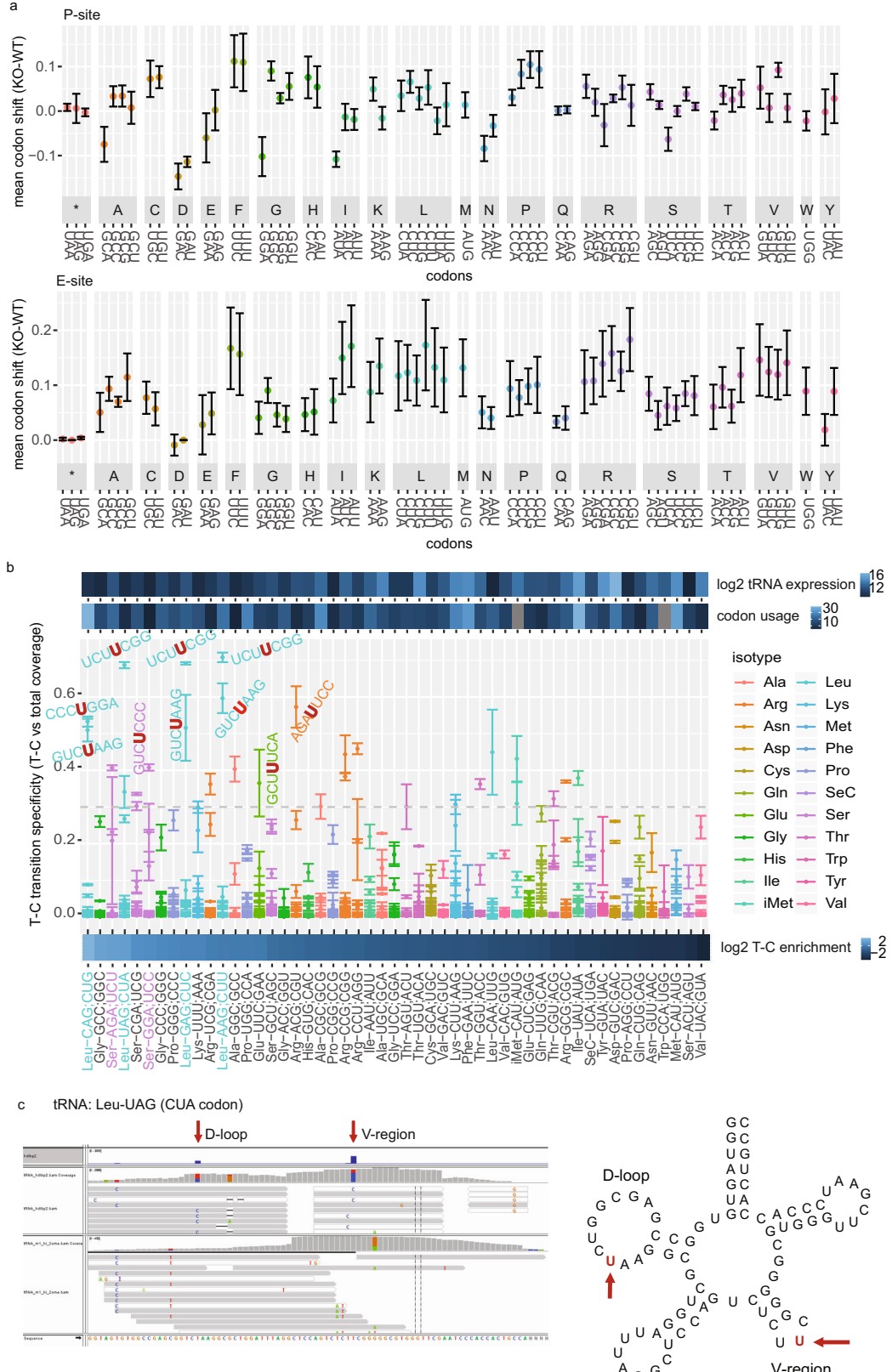

translation can be carried out in the presence of ER folding machinery and aggregation quality control. In the future, the sequence of HDLBP binding events to mRNA, rRNA, and tRNA during different stages of translation should be addressed.

In the absence of HDLBP, we also detected increased ribosome occupancy at P- and E-sites. Since this increase was modest and not

statistically significant, it may be a consequence of the indirect effects of HDLBP KO. Nevertheless, several studies suggest that E-site may act as a sensor for ribosome elongation kinetics[57] and that E-site occupancy may directly influence translation fidelity and ribosome translocation[58,59], which could be another possible mechanism of how HDLBP promotes translation elongation.

**Fig. 6 HDLBP crosslinks to tRNAs decoding CU/UU-containing codons. a** Difference in codon frequencies in the P-site (top) and E-site codons (bottom) in HDLBP KO vs. WT. Mean codon shift was calculated for four replicates (mean ± standard deviation is shown). **b** Enrichment of tRNAs in HDLBP PAR-CLIP and their binding sites. T-C transitions in tRNAs were normalized to total tRNA abundance and ranked from highest to the lowest value (left to right). For each T-C transition, we displayed its transition specificity (T-C transition vs. total read coverage). Mean values of two PAR-CLIP biological replicates ± SD are depicted. Total log2-transformed tRNA abundance and codon usage are also shown (top). **c** (Left) Browser representation of alignment to tRNA Leu-UAG. T-C transitions in the D-loop and V-region are indicated for the HDLBP PAR-CLIP dataset. The second track shows coverage in the total RNA sample. (Right) HDLBP crosslinked uridines are indicated with respect to secondary tRNA structure.

Overall, our results point to a general function for HDLBP in ER translation. Although this is obviously important for every cell, we expect that this phenotype is most prominent in specialized secretory cell types (e.g., fibroblasts, pancreatic and immune cells). In support of this view, HDLBP mRNA expression, and thus likely HDLBP protein level is substantially elevated in secretory cells (Supplementary Fig. 7g), most prominently in fibroblasts, which are key to orchestrating the extracellular landscape by producing and remodeling the extracellular matrix. In addition, HDLBP misregulation gives rise to far-reaching consequences and modulation of disease phenotypes, such as impaired viral replication[24], atherosclerotic plaque formation[35], and autism[60]. Finally, the results presented in this study highlight the striking involvement of HDLBP in lung tumor cells during tumor progression and suggest that therapeutic interventions targeting HDLBP may represent a previously unrecognized strategy for inhibiting lung tumor growth or other malignancies. In the future, further implications of the regulatory role of HDLBP for tumor biology need to be explored.

## Methods

**Cell lines and culture conditions**. HEK293 Flp-In T-REx (HEK293) (Thermo Fisher Scientific), HEK293 stable cell lines, and A549 cells were cultured in standard Dulbecco's modified Eagle´s medium (DMEM, Thermo Fisher Scientific) supplemented with 10% fetal bovine serum (FBS, Sigma-Aldrich) and 1% L-glutamine (200 mM, Thermo Fisher Scientific).

Stable HEK293 cell lines expressing HDLBP FLAG/HA or BirA*-FLAG-HDLBP were generated by hygromycin selection[61]. Induction of the stable cell lines was achieved by adding 1 μg/ml of doxycycline to the culture medium and incubation for 16 h.

HEK293 and A549 HDLBP knockout cell lines were produced using the Edit-R CRISPR-Cas9 Gene Engineering kit (Dharmacon) according to the manufacturer´s instructions. Briefly, transfections of synthetic tracrRNA (U-002000-05), hCMV-PuroR-Cas9 (U-005100-120), and pre-designed HDLBP crRNA (either guide 1 (CR-019956-01-0005) or guide 2 (CR-019956-04-0005)) or a non-targeting control (U-007501-05) were carried out using DharmaFECT Duo transfection reagent (Dharmacon, T2010-01) in a 12-well plate. After 2 days cells were reseeded to a 10 cm dish and treated with puromycin (2 μg/ml for HEK293 cells and 1 μg/ml for A549 cells). The surviving colonies were picked and Western analysis was performed.

A stable A549 cell line expressing HDLBP was generated by co-transfection of PB-TAG-ERP2-HDLBP and pCMV-hyPBase[62] in a 12-well plate using Lipofectamine 2000 (Thermo Fisher Scientific) according to manufacturer´s instructions. A puromycin selection was carried out and induction of the stable cell line was achieved by adding 1 μg/ml of doxycycline to the culture medium.

**Plasmids**. Vector pDONR221 carrying the HDLBP coding sequence was obtained from the hORFeome V5.1 collection and recombined into pFRT/TO/FLAG/HA-DEST (Addgene ID: 26360), pDEST5-BirA-FLAG-N-term-pcDNA5-FRT-TO[63], and PB-TAG-ERP2 (Addgene ID: 80479) using the Gateway LR Clonase II (Thermo Fischer Scientific) according to the manufacturer´s protocol. To purify recombinant full-length HDLBP, we amplified HDLBP from pENTR221_HDLBP with the primers HDLBP_fwd_SphI and HDLBP_end_rev_NotI (see Oligonucleotides) and ligation into SphI and NotI restriction sites in pQLinkG[64]. To purify the HDLBP protein variant A, we amplified variant A from pENTR221 HDLBP with the primers HDLBP_A_fwd_BamHI and HDLBP_A_rev_NotI (see Oligonucleotides) and ligation into BamHI and NotI restriction sites in pQLinkG. To purify the HDLBP protein variant B, we amplified variant B from pENTR221 HDLBP with the primers HDLBP_B_fwd_SphI and HDLBP_kh9_rev_NotI (see Oligonucleotides) and ligation into SphI and NotI restriction sites in pQLinkG. To purify the HDLBP protein variant C, we amplified variant C from pENTR221 HDLBP with the

primers HDLBP_C_fwd_BglII and HDLBP_end_rev_NotI (see Oligonucleotides) and ligation into BglII and NotI restriction sites in pQLinkG. To purify the HDLBP protein variant D, we amplified variant D from pENTR221 HDLBP with the primers HDLBP_D_fwd_BamHI and HDLBP_kh9_rev_NotI (see Oligonucleotides) and ligation into BamHI and NotI restriction sites in pQLinkG. For the SEAP secretion assays, we transfected pEZX-GA01 (GeneCopoeia) and additionally pFRTpsiCHECK containing Renilla luciferase and firefly luciferase. For the SEAP rescue secretion assays we replaced Renilla luciferase in pFRTpsiCHECK with SEAP luciferase by amplification from pEZX-GA01 with the primers SEAP_fwd and SEAP_rev (see Oligonucleotides) and ligation into NheI and NotI restriction sites. To carry out the Gaussia luciferase secretion assay, we replaced Renilla luciferase in pFRTpsiCHECK with *Gaussia* luciferase by amplification from pEZX-GA01 with the primers Gaussia_fwd and Gaussia_rev (see Oligonucleotides) and ligation into NheI and NotI restriction sites. Vectors have been submitted to Addgene.

**BioID proximity ligation assay**. The BioID proximity ligation assay was performed as described before[63] with minor modifications. Stable cell lines expressing BirA-FLAG/HDLBP (four 15 cm dishes per replicate) were incubated in the absence or presence of 1 μg/ml doxycycline for 24 h. Next, 250 μM biotin was added for 3.5 h. Cells were washed four times with PBS, harvested, snap-frozen in liquid nitrogen, and stored at −80 °C. Cell pellets were incubated with 3 ml RIPA buffer (50 mM Tris-HCl [pH 7.5], 150 mM NaCl, 1% IGEPAL CA-630, 1 mM EDTA, 1 mM EGTA, 0.1% SDS, 0.5% sodium deoxycholate, complete EDTA-free protease inhibitor cocktail [Roche]) per replicate for 10 min on ice. The cell lysates were passed eight times through a 21 G needle and sonicated (six times 5-s-pulses at 30% amplitude). About 250 U benzonase (Merck Millipore) was added per replicate for 1 h on the ice at slow agitation. Cell lysates were cleared (15,000 × g, 15 min, 4 °C) and filtered through a 5 μm Supor membrane. To wash streptavidin sepharose (GE Healthcare, 17-5113-01) the suspension was centrifuged at 400 × g for 1 min, the supernatant was removed and 1 ml RIPA buffer (without 0.5% sodium deoxycholate and without complete EDTA-free protease inhibitor cocktail) was added. The wash step was repeated two times. The RIPA buffer was removed and 40 μl per replicate of sepharose was added to the sample. After 3 h incubation at 4 °C on a rotating wheel the sepharose was washed twice with RIPA buffer, twice with TAP lysis buffer (50 mM HEPES-KOH [pH 8.0], 100 mM KCl, 10% glycerol, 2 mM EDTA, 0.1% IGEPAL CA-630) and three times with 50 mM ammonium bicarbonate. About 90% of the sample was stored at −80 °C and further processed for mass spectrometry.

**Mass spectrometry BioID**. Beads were resuspended in 200 μl of 50 mM ammonium bicarbonate containing 2 μl trypsin (Promega, V511A). The samples were incubated for 16 h in a Thermomixer (Eppendorf) at 37 °C and 750 rpm shaking. Afterwards, 1 μg of trypsin was again added and the samples were incubated further for 2 h. Samples were then centrifuged at 400 × g for 2 min and the supernatant was transferred to a new vial. To ensure complete beads removal, the samples were centrifuged at 16,000 × g for 10 min and the supernatants were transferred to a new vial containing 5 μl of trifluoroacetic acid. Each sample was loaded on two StageTips[65] for desalting. Eluates for each sample were pooled together prior to MS analysis.

For all the samples, 5 μl were injected in duplicate on an LC-MS/MS system (EkspertNanoLC 415 [Eksigent] coupled to Q-Exactive HF [Thermo]), using a 240 min gradient ranging from 2 to 45% of solvent B (80% acetonitrile, 0.1% formic acid; solvent A = 5% acetonitrile, 0.1% formic acid). For the chromatographic separation, a 30 cm long capillary (75 μm inner diameter) was packed with 1.8-micron C18 beads (Reprosil-AQ, Dr. Maisch). One end of the capillary nanospray tip was generated using a laser puller (P-2000 Laser-Based Micropipette Puller, Sutter Instruments), allowing fretless packing. The nanospray source was operated with a spay voltage of 2.3 kV and an ion transfer tube temperature of 260°. Data were acquired in data-dependent mode, with a top ten method (one survey MS scan with resolution 60,000 at m/z 200, followed by up to ten MS/MS scans on the most intense ions, resolution 15000, intensity threshold 5000). Once selected for fragmentation, ions were excluded from further selection for 30 s, in order to increase new sequencing events.

Raw data were analyzed using the MaxQuant proteomics pipeline (v1.5.3.30 and v1.5.8.3) and the built-in Andromeda search engine[66] with the Uniprot Human database. Carbamidomethylation of cysteines was chosen as fixed

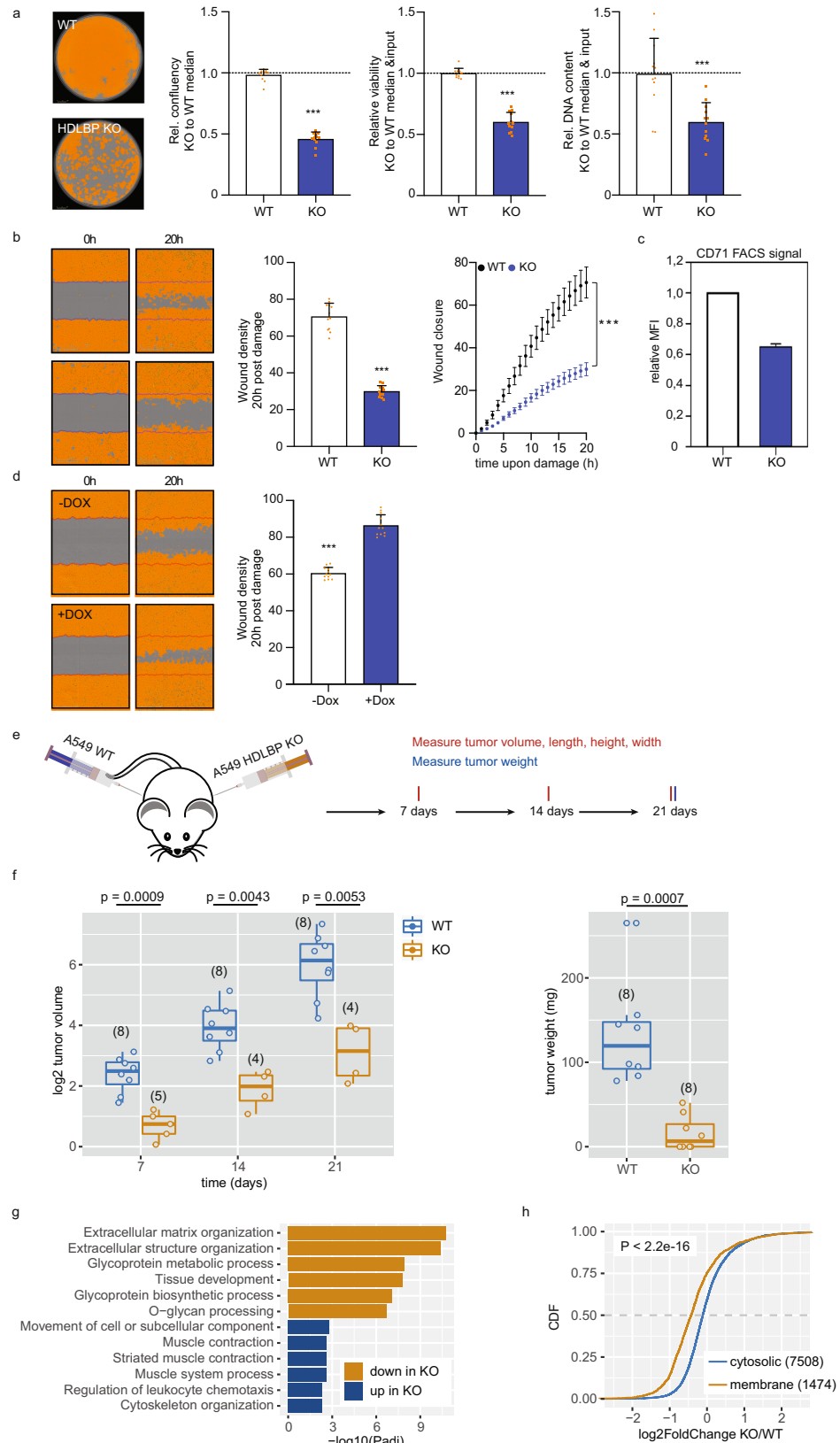

modification, oxidation of methionine, and acetylation of N-terminus were chosen as variable modifications. The search engine peptide assignments were filtered at 1% FDR and the feature match between runs was not enabled; other parameters were left as default.

Mass spectrometry proteomics data have been deposited to ProteomeXchange Consortium (http://proteomecentral.proteomexchange.org) via the PRIDE[67] partner repository with the dataset identifier PXD018313.

**Western analysis**. Cell pellets were lysed directly in Laemmli buffer, sonicated (5-s-pulse at 80% amplitude), and centrifuged (10,000 × g, 10 min, 4 °C). Before resolving the proteins by SDS-PAGE samples were boiled for 3 min (95 °C). For transferring the proteins to a nitrocellulose membrane (Whatman) a semi-dry blotting apparatus (20 V for 1 h) was used. The membranes were blocked with 5% non-fat milk for 1 h and incubated with the primary antibody overnight (anti-HDLBP (Abcam, ab109324; dilution 1:10000), anti-beta-Tubulin (Sigma, T8328;

**Fig. 7 HDLBP is required for tumor xenograft growth and formation. a** Relative confluence, viability, and DNA content of A549 parental or HDLBP KO cells. Representative confluence masks are shown (left panel). Unpaired *t*-test was used to test for significance (****p* < 0.001). **b** Wound healing of WT or HDLBP KO cells. The wound closure was monitored for 20 h upon damage (right panel). The wound density was calculated 20 h post damage (middle panel). Representative confluence masks are shown (left panel). Unpaired *t*-test (left) and two-way ANOVA (right) were used to test for significance (both ****p* < 0.001). **c** CD71 surface expression of WT or HDLBP KO cells. The mean fluorescent intensity (MFI) of CD71 was determined by flow cytometry experiments. **d** Wound healing of A549 cells stably transfected with doxycycline-inducible HDLBP overexpression (see Fig. 5g). The wound density was calculated 20 h post damage (right panel). Representative confluence masks are shown (left panel). Unpaired *t*-test was used to test for significance (****p* < 0.001). **e** Schematic of mouse injection experiment. WT or HDLBO KO A549 cells were injected into the right and left flanks of athymic nude mice. Tumor formation was followed in eight mice over 21 days post-injection. **f** Tumor volume (in mm³) was quantified 7, 14, and 21 days post-injection (left panel). Box plots of log2-transformed tumor volume formed from KO or WT cells are shown. An unpaired two-sided *t*-test was used for comparisons. The number of palpable tumors per group is given in parentheses. Tumor weight was quantified 21 days post-injection (right panel). Box plots of tumor weight (in mg) formed from KO or WT cells are shown. An unpaired two-sided *t*-test was used for comparisons. In case of the absence of a tumor, weight was plotted as 0. The lower and upper hinges of box plots correspond to the 25th and 75th percentiles, respectively. Upper and lower whiskers extend from the hinge to the largest or smallest value no further than the 1.5× interquartile range from the hinge, respectively. Center lines of box plots depict the median values. **g** Gene Ontology enrichment analysis of down and upregulated mRNAs in tumors from HDLBP KO cells. Adjusted *p* values for the top six enriched categories are shown. **h** Differences in fold change (HDLBP KO vs. WT tumors) were compared between membrane-bound and cytosolic mRNAs, as determined in the HEK293 RNA-seq fractionation experiment. A two-sided Wilcoxon rank-sum test was used to test for significance. **a–d** Data were presented as mean values ± SD. **a–h** Source data are provided as a Source Data file.

diluted 1:5000), anti-HA (Covance, MMS-101P-1000; diluted 1:5000), anti-BCAP31 (Proteintech, 11200-1-AP; diluted 1:2000), anti-RPS6 (Cell Signaling, #2217; diluted 1:1000), and anti-RPL7 (Abcam, ab72550; diluted 1:5000). After washing the membranes three times in TBST (150 mM NaCl, 20 mM Tris-HCl (pH 7.5), 0,1% Tween 20) the membranes were incubated with 1:4000 dilution of horseradish peroxidase-conjugated secondary antibodies (goat anti-rabbit immunoglobulins/HRP—Agilent cat# P044801-2 and goat anti-mouse immunoglobulins/HRP—Agilent cat# P044701-2) for 2 h. The membranes were washed three times with TBST, bands were visualized with ECL detection reagent (GE Healthcare), and imaged with an ImageQuant LAS 4000 imaging system or an Amersham Imager 680 imaging system (GE Healthcare).

**Cell fractionation**. Cell fractionation by sequential detergent extraction was performed as previously described[39] with minor modifications. HEK293 and HEK293 HDLBP knockout cell lines (one 15 cm dish per replicate) were grown to ~90% confluency. Cells were washed with PBS. All further steps were carried out on ice using ice-cold reagents and cells were always pelleted at 3000 × g for 5 min at 4 °C. First, PBS containing 50 µg/ml cycloheximide was added for 10 min. In the meantime, cells were scraped using a rubber policeman. After pelleting, the cells were resuspended with 500 µl permeabilization buffer (110 mM KOAc, 25 mM K-HEPES [pH 7. 2], 2.5 mM Mg(OAc)₂, 1 mM EGTA, 0.015% digitonin, 1 mM DTT, 50 µg/ml cycloheximide, complete EDTA-free protease inhibitor cocktail [Roche], 40 U/mL SUPERaseIn [Thermo Fisher Scientifc]) per sample and incubated for 15 min at 4 °C on a rotating wheel. After centrifugation, the supernatant was collected as the cytosolic fraction. To wash the pellet, 5 ml of washing buffer (110 mM KOAc, 25 mM K-HEPES [pH 7.2], 2.5 mM Mg(OAc)₂, 1 mM EGTA, 0.004% digitonin, 1 mM DTT, 50 µg/ml cycloheximide) was used. After pelleting, 500 µl lysis buffer (400 mM KOAc, 25 mM K-HEPES [pH 7.2], 15 mM Mg(OAc)₂, 0.5% IGEPAL CA-630, 1 mM DTT, 50 µg/ml cycloheximide, complete EDTA-free protease inhibitor cocktail, 40 U/mL SUPERase•In) per sample was added for 5 min. After centrifugation, the supernatant was collected as the membrane fraction. Fractions were clarified by centrifugation at 7500 × g for 10 min at 4 °C. About 250 µl of each sample was collected and RNA was isolated using Trizol LS (Thermo Fisher Scientific) in combination with RNA Clean & Concentrator-25 kit (Zymo Research) for RNA sequencing (see RNA sequencing library preparation).

**PAR-CLIP**. PAR-CLIP was performed as described previously[68] with minor modifications. Stable cell lines expressing HDLBP FLAG/HA were incubated with 100 µM 4SU for 16 h (20 dishes (15 cm) per replicate). Cells were UV crosslinked (365 nm, 0.15 J/cm²), harvested, snap-frozen, and stored at −80 °C. Cell pellets were incubated with three cell pellet volumes of lysis buffer (50 mM Tris-HCl [pH 7.4], 100 mM NaCl, 1% IGEPAL CA-630, 0.1% SDS, 0.5% sodium deoxycholate, complete EDTA-free protease inhibitor cocktail [Roche]) for 30 min on ice. Cell lysates were cleared (13,000 rpm, 15 min, 4 °C), filtered and RNase T1 was added at a final concentration of 1 U/µl for 10 min at 22 °C. About 20 µl beads (Dynabeads Protein G, Thermo Fisher Scientific) per 1 ml lysis buffer were washed twice with lysis buffer and incubated for 1 h in the presence of 0.25 µg/µl FLAG antibody (Sigma-Aldrich, F1804). Afterwards, beads were washed with lysis buffer twice and added to the sample. After 2 h incubation at 4 °C on a rotating wheel, the beads were washed with IP wash buffer (50 mM HEPES-KOH [pH 7.5], 300 mM KCl, 0.05% IGEPAL CA-630, 0.5 mM DTT, complete EDTA-free protease inhibitor cocktail) three times and a second RNase T1 treatment was carried out. For replicate 1 a final concentration of 1 U/µl and for replicate 2 a final concentration of 15 U/µl was used for 10 min at 22 °C. Subsequently, beads were washed with

high-salt wash buffer (50 mM Tris-HCl [pH 7.4], 1 M NaCl, 1% IGEPAL CA-630, 0.1% SDS, 0.5% sodium deoxycholate, 1 mM EDTA, complete EDTA-free protease inhibitor cocktail [Roche]) three times and two times with polynucleotide kinase buffer (50 mM Tris-HCl [pH 7.5], 50 mM NaCl, 10 mM MgCl₂, 5 mM DTT). Samples were radiolabeled using T4 polynucleotide kinase (NEB) and resolved on a Novex 4–12% BisTris gel (Thermo Fischer Scientific). The protein–RNA complexes were transferred to a nitrocellulose membrane (Whatman), exposed to a phosphorimager screen for 30 min, and excised at ~160 kDa. After proteinase K (Roche, 40 min at 50 °C) digestion the RNA was extracted by phenol-chloroform treatment and precipitation. To prepare sequencing libraries the RNA was firstly ligated to 3′ adapter 4N-RA3 (see Oligonucleotides) and gel-purified using a 15% denaturing urea-PAGE gel (Carl Roth). Next, the 5′ adapter OR5-4N (see Oligonucleotides) was ligated and gel-purified. The RNA was reverse transcribed and PCR-amplified by Phusion High-Fidelity DNA polymerase (Thermo Fischer Scientific) (see Oligonucleotides). The cDNA was visualized on a 2.5% agarose gel, a 140–160 bp sized fragment was excised and purified by a Zymoclean Gel DNA Recovery kit (Zymo Research). Next-generation sequencing was carried out on a HiSeq 2500 Illumina instrument (1 × 51 + 7 cycles).

**Ribosome profiling library preparation**. Ribosome profiling was performed as described previously[69] with minor modifications. HEK293 and HEK293 HDLBP knockout cell lines (one 10 cm dish per replicate) were grown to ~90% confluency. Cells were washed with ice-cold PBS containing 100 µg/ml cycloheximide. The PBS was thoroughly removed and plates were put on liquid nitrogen for 10 s and subsequently on ice. About 400 µl mammalian polysome buffer (20 mM Tris-HCl [pH 7.4], 150 mM NaCl, 5 mM MgCl₂, 1 mM DTT, 100 µg/ml cycloheximide, 1% Triton X-100, and 25 U/mL TurboDNase [Thermo Fisher Scientific]) per plate was added. Cells were scraped, collected, and the lysates were passed ten times through a 26 G needle. After clearing the cell lysates by centrifugation (20,000 × g, 10 min, 4 °C) 120 µl cell lysate aliquots were snap-frozen and stored at −80 °C. One aliquot of cell lysate was used for RNA sequencing (see RNA sequencing library preparation) and therefore RNA was isolated using Trizol LS (Thermo Fisher Scientific) in combination with RNA Clean & Concentrator-25 kit (Zymo Research). Another aliquot of cell lysate was used to isolate the ribosome-protected fragments by adding 300 U RNase I (Thermo Fisher Scientific) for 45 min at room temperature at slow agitation. Meanwhile, MicroSpin S-400 HR Columns (GE Healthcare) were equilibrated by adding regularly cold mammalian polysome buffer (without DTT, cycloheximide, Triton X-100, TurboDNase) to the columns. The columns were centrifuged (600 × g, 4 min, 4 °C). About 100 U SUPERaseIn (Thermo Fisher Scientific) per sample were added, mixed, and subsequently, the cell lysates were applied dropwise to the columns (100 µl cell lysate per column). The columns were centrifuged (600 × g, 2 min, 4 °C) and the flow-through was collected. RNA was isolated using Trizol LS (Thermo Fisher Scientific) in combination with RNA Clean & Concentrator-25 kit (Zymo Research). The ribosome-protected fragments were then depleted with the RiboZero Kit (Illumina) according to the manufacturer´s protocol by using 5 µg RNA as input. The remaining RNA was separated on a 17% denaturing urea-PAGE gel (Carl Roth) and RNA fragments in the range from 27 to 30 nt were excised (defined by markers (see Oligonucleotides)). Sequencing libraries were generated as described in the PAR-CLIP section. Next-generation sequencing was carried out on a HiSeq 2500 Illumina instrument (1 × 51 + 7 cycles).

**Pulsed SILAC**. HEK293 and HEK293 HDLBP knockout cell lines were cultured for at least three passages in arginine- and lysine-free DMEM (Life Technologies)

containing 10% dialyzed FBS (Pan-Biotech), 1% glutamax (Life Technologies), 1% sodium pyruvate (Life Technologies), and "light" form amino acids 0.2 mM L-arginine (Sigma-Aldrich) and 0.8 mM L-lysine (Sigma-Aldrich). Cells were seeded in six-well plates (450,000 cells per well). After 48 h the "light" form medium was removed. It was replaced by either medium containing the "medium" form amino acids (L-[13C6]-arginine (Sigma-Aldrich), L-[2H4]-lysine (Cambridge Isotope Laboratories)) or "heavy" form amino acids (L-[13C6,15N4]-arginine (Sigma-Aldrich), L-[13C6,15N2]-lysine (Cambridge Isotope Laboratories)). Cells were fractionated after 2 h or 4 h. Cell fractionation was carried out as described above with reduced volumes: six wells were washed with cold PBS. About 544 μl PBS containing 50 μg/ml cycloheximide was added for 10 min. In the meantime, cells were scraped and one 6-well HEK293 and one 6-well HEK293 HDLBP knockout cells were combined in an Eppendorf tube. Downstream 100 μl of permeabilization buffer, 544 μl of wash buffer, and 100 μl of lysis buffer were used. After clarification of the fractions 90 μl sample was recovered and 810 μl pure EtOH was added and samples were submitted to mass spectrometry.

**Mass spectrometry pulsed SILAC.** Protein samples were resuspended in 6 M urea, 2 M Thiourea, and 10 mM HEPES pH 8 solution. Proteins were reduced with 10 mM DTT and alkylated with 55 mM iodoacetamide at room temperature. For lysis, proteins were incubated with 1% (w/w) lysyl endopeptidase (Wako) at room temperature for 3 h; diluted with 50 mM ammonium bicarbonate solution for a final urea concentration of 2 M, and incubated with 1% (w/w) trypsin (Promega) under constant agitation at room temperature for 16 h. Peptides were acidified with 1% (v/v) trifluoroacetic acid and desalted with C18 StageTips[65]. Prior to LC-MS/MS analysis, peptides were eluted with 50% acetonitrile 0.1% formic acid, dried, and resuspended in 3% acetonitrile, 0,1% formic acid (Buffer A). The peptide concentration was measured based on 280 nm UV light absorbance.

Reversed-phase liquid chromatography was performed employing an EASY nLC II (Thermo Fisher Scientific) using self-made C18 microcolumns (75 μm ID, packed with ReproSil-Pur C18-AQ 1.9-μm resin, Dr. Maisch, Germany) connected online to the electrospray ion source (Proxeon, Denmark) of an Orbitrap HF-X or an OrbitrapExploris 480 mass spectrometer with the FAIMS module installed in application mode "Peptide" (Thermo Fisher Scientific). Peptides were eluted at a flow rate of 250 nL/min over 2 or 4 h with a stepwise increasing gradient of 4.74 to 81.3% acetonitrile in constant 0.1% formic acid. Settings for data-dependent analysis on Orbitrap HF-X were: positive polarity, full scan (resolution, 60,000; m/z, 350–1800, AGC target, 3e6; injection time, 10 ms) followed by top20 MS/MS scans with higher-energy collisional dissociation (resolution, 15,000; m/z, 200–2000; AGC target, 1e5. injection time, 22 ms; isolation width, 1.3 m/z; normalized collision energy, 26). Settings for data-dependent analysis on Exploris were: positive polarity, 1 s cycle time, full scan (resolution, 60,000; m/z, 350–1800, AGC target, 300%; injection time, 30 ms; FAIMS CV, −40, −55, or −70 mV) followed by MS/MS scans with higher-energy collisional dissociation (resolution, 7500; m/z, 200–2000; AGC target, 100%. injection time, 25 ms; isolation width, 1.3 m/z; normalized collision energy, 28). Ions with an unassigned charge state, singly charged ions, and ions with a charge state higher than five were rejected. Former target ions selected for MS/MS were dynamically excluded.

All samples were measured in technical duplicates. Raw files were analyzed together with MaxQuant software (v1.6.0.1)[70] with default parameters. Files measured by Q-exactive HF-X or Exploris machines were grouped in the same experimental group, while technical and biological replicates were kept separated. Briefly, search parameters included two missed cleavage sites, cysteine carbamidomethyl as fixed modification, methionine oxidation, protein N-terminal acetylation, and asparagine or glutamine deamidation (only identification) as variable modifications. Triple multiplicity was used for the search of light (Lys0 and Arg0), medium-heavy (Lys4 and Arg6), and heavy (Lys8 and Arg10) peptides. Peptide mass tolerance was 20 and 4.5 ppm for the first and main search, respectively. A database search was performed with Andromeda embedded in MaxQuant[66] against UniProt/Swiss-Prot human database (downloaded on January 2019) with common contaminant sequences provided by MaxQuant. The false discovery rate (FDR) was set to 1% at peptide spectrum match (PSM) and protein levels. The minimum peptide count required for protein quantification was set to two. The match between runs was turned on. The search was performed with Requantify turned on, and again with the option turned off.

For analysis, potential contaminants, reverse database hits, and peptides only identified by modification were excluded. Unscrupulous ratios, defined as a SILAC pair quantified from both requantified intensities, were removed from further analysis. These were rare and constituted only a small fraction of requantified SILAC ratios. MaxQuant normalized SILAC ratios were used for analysis. For average calculations, only proteins with one or more values in at least one replicate from both forward and reverse SILAC label experiments were accepted. For the forward experiment, the heavy SILAC label corresponded to the HDLBP KO cells, while the medium SILAC label corresponded to WT cells. In the reverse experiment, the labels were switched. We noticed that the samples from the membrane fractions after 4 h of gradient fractionation contained much higher iBAQ values for SP and TM-containing proteins. Therefore, we only used membrane fraction results after 4 h for further analysis. Protein names from the MaxQuant output were mapped to the RNA-seq fractionation data table to obtain the same classification as in other analyses. Distributions of H/M SILAC ratios

between different classes of proteins were compared by cumulative density function and significance was evaluated by the Wilcoxon rank-sum test.

Mass spectrometry proteomics data have been deposited to ProteomeXchange Consortium (http://proteomecentral.proteomexchange.org) via the PRIDE partner repository with the dataset identifier PXD018316.

**Gaussia luciferase and SEAP assays.** HEK293 and HEK293 HDLBP knockout cell lines were seeded in six-well plates (500,000 cells per well). After 24 h cells were transfected using Fugene6 reagent (Promega) with plasmids expressing either Gaussia luciferase (Gluc) and Firefly luciferase (Fluc) or secreted alkaline phosphatase (SEAP) and Fluc according to the manufacturer´s protocol. Forty-eight hours after transfection cells were split into white bottom 96-well plates (Thermo Fisher Scientific, 136101) (70,000 cells per well). Twenty-four hours later 20 μl medium per well was transferred to a new white bottom 96-well plate. The Gluc signal was measured using the Gaussia Luciferase Flash Assay Kit (Thermo Fisher Scientific, PI16159) and the SEAP signal was measured using the NovaBrightPhospha-Light EXP Assay Kit (Thermo Fisher Scientific, N10577) according to the manufacturer´s protocol. The Fluc activity was measured in the remaining medium and cells with the Firefly Luc One-Step Glow Assay Kit (Thermo Fisher Scientific, PI16197). Gluc and SEAP signal was normalized to the Fluc signal and each experiment was carried out using five technical replicates. For SEAP rescue experiments (Fig. 5g) we transfected HEK293 and HEK293 HDLBP KO2 cells with SEAP and Fluc and additionally with pFRT/TO/FLAG/HA-HDLBP. In addition, we transfected A549, stable A549 overexpressing HDLBP and A549 HDLBP KO with SEAP and Fluc. Twenty-four hours after transfection cells were split into white bottom 96-well plates and the rest of the experiment was carried out as described above. Each rescue experiment was carried out using at least five technical replicates and the average of five biological experiments is shown.

**RNA co-immunoprecipitation.** Stable cell lines expressing HDLBP FLAG/HA or BirA*-FLAG-HDLBP were scraped, centrifuged in a capped syringe, and frozen by pressing the cells through the syringe directly in liquid nitrogen. Frozen cells were grinded, aliquoted as powder, and stored at −80 °C. About 200 mg cryopowder were incubated with 1 ml of lysis buffer (50 mM Tris-HCl [pH 7.4], 150 mM NaCl, 0.5% IGEPAL CA-630, complete EDTA-free protease inhibitor cocktail [Roche]) for 30 min on ice and sonicated (5 s at 20% amplitude). Cell lysates were cleared (13,000 rpm, 15 min, 4 °C). About 25 μl beads (Dynabeads Protein G, Thermo Fisher Scientific) per 1 ml lysis buffer were washed twice with lysis buffer and coated for 1 h with 0.25 μg/μl FLAG antibody (Sigma-Aldrich, F1804) or with 0.25 μg/μl IgG antibody (Sigma-Aldrich, M5284). Afterwards, beads were washed with lysis buffer twice and added to the sample. After 2 h incubation at 4 °C on a rotating wheel, the beads were washed five times with lysis buffer, and subsequently, Laemmli buffer was added for Western Blot analysis or Trizol LS (Thermo Fisher Scientific) for RNA isolation. After RNA isolation DNase (NEB, M0303) treatment and reverse transcription using SuperScript III Reverse Transcriptase (Thermo Fisher Scientific, 18080093) was carried out according to the manufacturer's protocol. Real-time PCR was performed with SYBR Green PCR Master Mix (Thermo Fisher Scientific, 4309155) and primers listed under oligonucleotides. Fold enrichment was calculated from Ct values detected in anti-FLAG and IgG control samples (2^(anti-Flag CT value—IgG control CT value)).

**Xenograft assay.** The permission for in vivo xenograft assays was granted by an ethical review committee (Landesverwaltungsamt Sachsen-Anhalt; protocol number: AZ42502-2-1625). Athymic nude mice (Strain: Crl:NU(NCr)-Foxn1[nu]) were obtained from Charles River. Animals were handled according to the local guidelines of the Martin-Luther-University Halle-Wittenberg. Subcutaneous xenograft assays were essentially performed as previously described[71]. In brief, A549 Ctrl and HDLBP KO cells were transduced with iRFP-encoding lentiviruses at 10 MOI (multiplicity of injection). For the subcutaneous injection into nude mice, $1 \times 10^6$ cells were harvested in media supplemented with 50% (v/v) matrigel (Sigma-Aldrich). Ctrl and KO cells were injected into the left and the right flanks of six-week-old female athymic nude mice ($n = 8$). Mice were held with access to chlorophyll-free food (Altromin C1086) ad libitum to reduce background noise in weekly iRFP imaging using a Pearl Trilogy Imaging System (LI-COR). The fluorescence intensity was determined by using the Image Studio software (LI-COR). Tumor volumes were measured and calculated according to the formula $0.52 \times L_1 \times L_2 \times L_3$. The mice were sacrificed, once the first tumor reached the termination criteria of a 1.5 cm diameter. Palpable tumors were excised, the weight was measured, the RNA was isolated and sequencing libraries were prepared using the Next Ultra Directional RNA Library Prep Kit (NEB). Next-generation sequencing was carried out on a HiSeq 4000 Illumina instrument ($1 \times 51 + 7$ cycles).

**Cell proliferation and migration assays.** For the determination of cell proliferation, $1 \times 10^3$ Ctrl or HDLBP KO A549 cells were seeded in 96-well plates. Cell confluency was monitored for 5 days using an IncuCyte S3 system (Sartorius) with 4× magnification for a whole well scan. Confluence masks were generated using the IncuCyte analysis software. Cell viability and DNA content were determined using

CellTiter Glo (Promega) supplemented with 1/2000 SYBR Green I (Thermo Fisher) according to the manufacturer's protocols. Luminescence and fluorescence intensities were measured in a GloMax microplate reader (Promega). For the scratch wound migration analysis, $2.5 \times 10^4$ Ctrl, HDLBP KO A549 cells and stable A549 cells expressing HDLBP (±doxycline) were seeded in a 96-well ImageLock plate (Sartorius) for 24 h. The wound areas were created on confluent cell monolayers using a 96-well WoundMaker (Sartorius). Wound healing was monitored using an IncuCyte S3 system at 10x magnification and 1 h intervals. Confluence and wound masks were generated and quantified using the IncuCyte analysis software.

**Flow cytometry**. To determine changes in CD71 expression and presentation, $2 \times 10^5$ HDLBP knockout and WT control cells were seeded in six-well plates and grown for 2 days. After harvesting using trypsin, cells were counted. For CD71 surface labeling, $3 \times 10^5$ cells were stained with anti-CD71 antibodies (human CD71, clone AC102, APC-conjugated; Miltenyi Biotec; RRI-D:AB_2660542; dilution 1:11) or isotype control (REA control antibodies, clone REA293, APC-conjugated; Miltenyi Biotec; RRID:AB_2733447; dilution 1:50) diluted in 1% BSA/PBS for 15 min at 4 °C. After washing with PBS, cells were analyzed by flow cytometry using a MACSQuant Analyzer (Core Facility Imaging; Martin-Luther University Halle-Wittenberg). Dead cells were excluded by propidium iodide (Miltenyi Biotec) staining. Mean fluorescence intensities of CD71 were determined upon background subtraction using the FlowJo analyses software. The experiment was carried out three times.

**PAR-CLIP data processing and analysis**. Reads were demultiplexed, stripped of the 3′ adapter sequence by Flexbar (v2.5), and collapsed to remove PCR duplicates. This was followed by trimming of four nucleotides from both the 5′ and 3′ end of the read using FASTX Toolkit v0.0.14. Next, the reads were aligned to the human genome (hg19 build) using BWA v0.7.15-r1140 and the previously published computational PAR-CLIP pipeline (v0.97a)[72,73] https://github.com/marvin-jens/clip_analysis. Briefly, read clusters were called from unique alignments and scored for characteristic T-C transitions. After false positive filtering (using antisense clusters as a decoy database and a false discovery rate of 0.05) the remaining clusters were written as bed files. Clusters obtained from each biological replicate were additionally filtered for reproducibility. We only considered those clusters that overlapped by at least 50% of their nucleotide length between replicates. In addition, we required that the positions of the highest T-C transition values per cluster were no more than ten nucleotides apart between two replicates. Each kept cluster was also required to have at least three or more mean T-C transitions calculated between replicates. To obtain gene-level binding information, we summed T-C transitions in reproducible clusters for each gene. We thus obtained the total number of crosslink positions within whole mRNAs, or within the CDS, 5′ UTR, or 3′ UTR. To correct for PAR-CLIP expression level bias, we divided the total number of crosslinks per gene by the corresponding TPM value as obtained by RSEM (v.1.2.2) from our RNA-seq experiment from unfractionated HEK293 cells (see RNA-seq description below). This provided us with PAR-CLIP enrichment values (Fig. 1d, f).

To assess the relative distribution of TC-transitions within the transcriptome and obtain accurate mapping to transcripts originating from repetitive genomic loci, reads were sequentially mapped to reference transcripts by Bowtie2 (v2.3.2) in the following order by retaining the unmapped reads from the previous to the next mapping step. We started with human pre-rRNA (GenBank U13369.1), followed by rRNA (GenBank NR_023363.1, NR_003285.2, NR_003287.2, NR_003286.2), snRNA, snoRNA, other ncRNA (all from Ensembl, including RN7SL), tRNA (GtRNAdb), mtDNA (GenBank AF347015.1), and finally the human genome (hg19, iGenomes). The last genome-mapping step was performed by the STAR aligner (v2.2.1) where only uniquely mapped reads (MAPQ = 255) were retained. Except for the tRNA mapping (see below for details), we retained reads that mapped with a Bowtie2 MAPQ = 20 or more and T-C transitions were extracted using row_mpile_coverage_plus_TC.pl script[74].

For transcriptome level analysis, remaining reads after mtDNA mapping were aligned to the transcriptome sequence (GTF annotation file Gencode v19) by the STAR aligner (v2.2.1) by retaining only the reads that uniquely mapped to the hg19 genome (parameters -outFilterMultimapNmax 1 -outFilterMismatchNmax 5 -outFilterMatchNmin 15 -alignSJoverhangMin 5 -seedSearchStartLmax 20 -outSJfilterOverhangMin 30 8 8 8 -quantMode TranscriptomeSAM). We considered only one transcript isoform per gene for further analysis. This filtering was carried out based on RSEM (v1.2.2) results obtained from our RNA-seq experiment from unfractionated HEK293 cells (see RNA-seq description below), so that only the most highly expressed transcript isoform for a given gene was retained. T-C transitions with transcriptome coordinates were obtained from the BAM file using the row_mpile_coverage_plus_TC.pl script[74] and written into a bed file. We used BedTools intersect to only retain reproducible T-C positions that were present in both replicates. Among those, we excluded the positions that were highly likely to be point mutations and not cross-linking sites by retaining only those that had transition specificity (positional T-C transition count vs. total read coverage) lower than 0.95. To assess the positional HDLBP crosslinks across 5′ UTR, CDS, and 3′ UTR (Fig. 2a), we used reproducible T-C positions and normalized them for library size to obtain T-C transitions per million. We included only transcripts where the total transcript T-C transitions per million were at least

five in both replicates. For each T-C position, we then obtained the relative positions within 5′ UTR, CDS, or 3′ UTR and divided the T-C per million value with the maximum T-C per million value of the corresponding transcript. For each position, we then averaged the scaled T-C per million values over all transcripts and plotted them with respect to their nucleotide positions within 5′ UTR, CDS, or 3′ UTR. Classification of membrane and cytosolic mRNAs was carried out according to the results of our RNA-seq fractionation experiment.

**RNA-seq library preparation, data processing, and analysis**. RNA sequencing libraries were prepared using the TruSeq Stranded mRNA kit (Ilumina) according to the manufacturer´s protocol by using 1 µg RNA as input. Next-generation sequencing was carried out on a HiSeq 4000 or NextSeq 500 Illumina instruments. Reads were demultiplexed and 3′ adapter sequences removed by Flexbar (v2.5). Read counts per gene and TPM values were obtained by RSEM (v.1.2.20)[75] using default parameters and Bowtie (v.1.1.2)[76] as transcriptome alignment program. Raw read counts were normalized using DESeq2[77] and pairwise comparisons between and within fractions were performed using standard parameters. Log2-transformed fold changes were obtained from the DESeq2 output. Membrane-to-cytosol enrichment was equal to the log2-transformed fold change between the membrane and cytosolic samples. To define membrane-bound and cytosolic mRNAs we used cutoffs of ≥1.5 and ≤0.5, respectively, for all mRNAs with TPM ≥10.

**Ribosome profiling data processing and analysis**. Reads were demultiplexed and adapter sequences were removed by Flexbar (2.5). Reads were then collapsed to remove PCR duplicates, followed by the removal of random nucleotides (four on both the 5′ and 3′ end of the reads) using fastx_trimmer (FASTX Toolkit 0.0.14). Reads aligning to rRNA sequences and other sources of contamination using a custom index were removed by Bowtie2 (v2.3.2) and the remaining sequences were aligned to the human transcriptome (hg19) using STAR aligner (2.5.3a) using GTF annotation file Gencode v19 by retaining only the reads that uniquely mapped to the hg19 genome (parameters -outFilterMultimapNmax 1 -outFilterMismatchNmax 5 -outFilterMatchNmin 15 -alignSJoverhangMin 5 -seedSearchStartLmax 20 -outSJfilterOverhangMin 30 8 8 8 -quantMode TranscriptomeSAM). Transcriptome BAM files were converted to the bed format using BedTools bamToBed (v2.26.0). Bed files were then input into riboWaltz (v1.1.0)[78], which we used for downstream quality control and P-site coverage analysis.

Quality control of ribosome profiling data were performed using riboWaltz, which outputs metaheatmaps of read coverage around the start and stop codons for all possible read lengths (Supplementary Fig. 5a). In addition, we used riboWaltz to calculate optimal P-site offset (13 nt) and to obtain P-site coverage per nucleotide. To include a P-site in the downstream metagene analysis, a minimal P-site nucleotide coverage of five was requested in at least one of the samples.

To perform the metagene analysis of P-site occupancy around the start and stop codons (Supplementary Fig. 5c), we excluded the P-site coverage corresponding to the first two and the last two codons of the coding sequence. The reason for this is the high read coverage at these positions due to ribosome initiation and termination, which would prevent meaningful normalization and mask the differences in the regions around the start and stop codons. P-site coverage per nucleotide was first normalized for library size and summed for each codon. Next, the codon coverage was scaled to the mean CDS coverage excluding the coverage at the extremities. We included all CDS that had total codon coverage of five or higher. For all interrogated codon positions within the CDS, the scaled P-site coverage was averaged. A rolling mean over 10 nt was used to smooth the signal in the final metagene plot.

To perform the metagene analysis of codon occupancy around targeting signals (Fig. 5c and Supplementary Fig. 5g), we summed the P-site coverage per codon. The obtained codon coverage was normalized to the mean coverage within codons 20–40 in each coding sequence, as described previously[9]. For all interrogated codon positions within the CDS (1–500), the scaled P-site coverage was then averaged over all well-quantified transcripts (mRNAs with TPM ≥10). A rolling mean over 5 nt was used to smooth the signal in the final metagene plot.

In order to calculate translation efficiencies per gene, we used RSEM (v1.2.2) which supplied us with read counts and TPM values per gene. Differences in translational efficiency, as well as in mRNA abundance and due to both effects (transcription and translation) were detected by DESeq2 (1.18.1) with an interaction term model as described previously[79]. Briefly, RPF read counts were normalized using the DESeq2 estimateSizeFactors function by considering all read counts. DESeq2 was run with default parameters. Log2-transformed fold changes for downstream comparisons were taken directly from the DESeq2 output.

For codon level analysis, we used the codon frequency analysis available within riboWaltz. Differences in codon frequency were obtained by subtracting the codon occupancy values between conditions and calculating standard deviations between replicates.

**Targeting signal annotations**. Signal peptide and transmembrane helix annotations were downloaded from Ensembl Biomart (http://grch37.ensembl.org/biomart/martview), which uses SignalP[80] and TMHMM[81] for annotation.

Annotations were downloaded in protein sequence coordinates and converted into transcript coordinates using Biostrings (v2.52.0, Bioconductor). Based on these annotations we defined tail-anchored proteins as those transmembrane domain-containing proteins lacking signal peptide, for which the first transmembrane helix started 50 or less amino acids from the C-terminus. We defined mitochondrial DNA-encoded proteins as those that contained an "MT-" prefix in their gene symbol. A list of mitochondrial proteins encoded in the nuclear DNA was obtained from Mitocarta 2[82].

**ncRNA-seq library preparation, data processing, and analysis**. For normalization of tRNA abundance in the HDLBP PAR-CLIP dataset we made use of the RNA-seq library preparation protocol based on small RNA cloning approaches[83–86]. Total RNA was extracted from HEK293 cells with Qiagen RNAeasy kit according to the manufacturer's instruction and 2 µg were fragmented in a total volume of 100 µl fragmentation solution (10 mM ZnOAc, 100 mM Tris buffer pH 8.0) in a thin-walled PCR tube at 95 °C for 55 s. Next, the samples were immediately cooled on ice and RNA was purified with RNA Clean and Concentrator kit (Zymo Research) according to the manufacturer's instructions. RNA was then treated with 10 U of calf intestine alkaline phosphatase in 1x CutSmart buffer (both from New England Biolabs) for 1 h at 37 °C. Next, the samples were immediately cooled on ice and RNA was purified with RNA Clean and Concentrator kit (Zymo Research) according to the manufacturer's instructions. RNA was then treated with 20 U of T4 polynucleotide kinase in 1× T4 PNK reaction buffer (both from New England Biolabs) supplemented with 0.1% (v/v) Triton-X-100 for 30 min at 37 °C. Next, the samples were immediately cooled on ice and RNA was purified with RNA Clean and Concentrator kit (Zymo Research) according to the manufacturer's instructions. The next step was 3′ adapter ligation, which was performed for 16 h at 16 °C in a total volume of 20 µl in the presence of 2 U of T4 RNA ligase 2 truncated K227Q, 1× RNA ligase buffer, 25% PEG-8000 (all from New England Biolabs) and 25 pmol of pre-adenylated 4N-RA3 adapter (see Oligonucleotides). Afterwards, RNA was separated on 10% Novex TBE Urea gels, and fragments in the range of 65–100 nt were excised and eluted in 300 mM NaCl overnight at 4 °C with 900 rpm agitation. RNA was then precipitated by the addition of 0.5 µl GlycoBlue co-precipitant, 3 volumes of absolute ethanol and incubation at −80 °C for at least 2 h. Next, precipitated RNA was reverse transcribed for 30 min at 37 °C in a total volume of 20 µl containing 1x buffer (25 mM Tris, pH 8.6, 30 mM NaCl, and 5 mM dithiothreitol), 4.5 pmol RTP primer (see Oligonucleotides), 4 mM each dNTP, 20 mM MgCl₂, 7.1% (v/v) dimethylsulfoxide, and 10 U of AMV reverse transcriptase (Promega). Afterwards, RNA was removed by RNase H (5 U) treatment for 30 min at 37 °C (New England Biolabs) and ssDNA was separated on 10% Novex TBE Urea gels. Fragments in the range of 65–100 nt were excised and eluted in 300 mM NaCl overnight at 4 °C with 900 rpm agitation. ssDNA was then precipitated by the addition of 0.5 µl GlycoBlue co-precipitant, 3 volumes of absolute ethanol, and incubation at −80 °C for at least 2 h. The next step was 5′ adapter ligation, which was performed for 16 h at 22 °C in a total volume of 20 µl. The reaction contained precipitated ssDNA, 60 U of high concentration T4 RNA ligase 1, 1× RNA ligase buffer, 22.5% PEG-8000, 1 mM ATP (all from New England Biolabs), and 50 pmol of 4N-SRC-cDNA adapter (see Oligonucleotides). ssDNA was purified with DNA Clean and Concentrator kit (Zymo Research) according to the manufacturer's ssDNA protocol. PCR amplification for the final library was performed according to the Truseq small RNA protocol (Illumina) using standard indexed PCR primers (RPIx) for multiplexing (Illumina). Typically, 14 cycles of amplification were required to obtain a sufficient amount of the final cDNA library in the size range of 150–220 bp (average size 180 bp). PCR reactions were separated on a 2.5% agarose gel and the final DNA library of the expected size range was excised and sequenced on a HiSeq 4000 sequencer (Illumina) using TruSeq 1 × 51 + 7 cycle kit.

Reads were demultiplexed and adapter sequences were removed by Flexbar (2.5). Reads were then collapsed to remove PCR duplicates, followed by the removal of random nucleotides (four on both the 5′ and 3′ end of the reads) using fastx_trimmer (FASTX Toolkit 0.0.14). To obtain accurate mapping results to transcripts originating from repetitive genomic loci, we used the same strategy as for PAR-CLIP analysis (see details above).

**Mapping to reference tRNAs and quantification**. Initially, we obtained mature tRNA fasta files from GtRNADb (http://gtrnadb.ucsc.edu/, hg19 build) and kept only one fasta sequence per tRNA molecule to avoid mapping non-uniquely due to tRNA gene duplications and pseudogenes. The tRNAs missing from this initial set were obtained from HEK293 Hydro-seq results[87], resulting in a custom-made FASTA tRNA reference that contained 58 unique tRNAs. Reads were mapped to this reference using Bowtie2 (v.2.3.2) and custom scripts using BedTools (v2.26.0) intersect and getfasta commands were used to obtain T-C transition counts per position and per tRNA. By inspection of the aligned reads in IGV and making use of the Modomics database (https://iimcb.genesilico.pl/modomics)[88], we noticed that T-C transitions were detected at dihydrouridine bases, which were present also in libraries prepared from total HEK293 RNA with no 4SU and UV treatment. We thus removed these positions from T-C transition counting by using a custom-made bed file of all dihydrouridine bases. To quantify the tRNA abundance in HEK293 cells we made use of our ncRNA-seq dataset from randomly fragmented total HEK293 RNA (see experimental details under ncRNA-seq). To calculate the

enrichment of tRNAs in HDLBP PAR-CLIP we first normalized T-C transition counts and read counts per tRNA for library size. Next, for each tRNA, T-C transitions counts were divided by the respective read count per tRNA from the ncRNA-seq samples to obtain the enrichment score. Transition specificity per T-C position within one tRNA was obtained by dividing the T-C transition count with total read coverage at the same nucleotide position. Codon usage information was obtained from https://www.kazusa.or.jp/codon/. The tRNA sequences, dihydrouridine positions along with the entire tRNA processing pipeline are available from GitHub https://github.com/mmilek/hdlbp_rev/tree/hdlbp-master/trna.

**BioID analysis**. We excluded those protein groups that were identified by MaxQuant as potential contaminants, and by reverse peptide sequences. We also requested that the number of razor unique peptides per protein group was at least 3. Fold change enrichment in HDLBP BioID was calculated by dividing the LFQ values from the doxycycline (Dox)-treated condition with untreated condition-derived LFQ values. Since we obtained three biological replicates (batches 1, 2, and 3) of the Dox-treated cells and only two biological replicates of the untreated controls (batches 2 and 3), we divided the Dox batch 1 with untreated batch 2 to finally obtain enrichment values for three replicates. In order for the protein group to be defined as enriched in the HDLBP BioID, we requested that its enrichment value was at least 3 and that the log2-transformed LFQ intensity in the Dox-treated condition was at least 27. For the protein groups, where the calculation of enrichment values was not possible due to the fact that LFQ intensity in the untreated condition equaled zero, we requested that the log2-transformed LFQ intensity in the Dox-treated condition was at least 27. These filtering criteria were applied separately for each biological replicate so that the protein group had to fulfill them in all three replicates to be defined as enriched. To rank the enriched proteins, we averaged their LFQ intensity in the Dox condition over three replicates. To visualize the top 60 enriched protein groups by mean LFQ, we also displayed the mean log2-transformed enrichment values. For those that did not have an enrichment value due to the fact that LFQ intensity in the untreated condition equaled zero, we used a maximum of the mean enrichment value and added a value of 2.

**K-mer enrichment and multivalency analysis**. For these analyses, we used the reproducible T-C transitions per million values from the transcriptome level analysis as described above in the PAR-CLIP section. To calculate the frequency of crosslinked k-mers (4–12 nt in length), all possible k-mers at any cross-linking position were counted for all membrane-bound and cytosolic transcripts in their 5′ UTRs, CDS, and 3′ UTRs. The frequency was calculated by dividing with the total detected k-mer number in the whole sample. K-mers were then ranked by their total frequency in the transcriptome and the top ten k-mer frequencies were displayed according to mRNA localization and transcript region (5′ UTR, CDS, and 3′ UTR).

To obtain k-mer-specific cross-linking quantification for each transcript and its regions, we summed the T-C transition counts for each possible k-mer within a CDS and 3′ UTR. To normalize this metric, we divided the values with the length of the respective CDS/3′ UTR and with the transcript expression level as quantified by the unfractionated mRNA-seq experiment (see above). The k-mers were then ranked according to the median of the log2-transformed normalized crosslinked k-mer signal.

Sequence analyses of differentially localized transcripts and their regions were carried out by calculating the k-mer frequency using the oligonucleotideFrequency function in Biostrings (v2.52.0, Bioconductor). We used fasta sequences of the same transcript subset as selected for PAR-CLIP and ribosome profiling analyses described above, i.e., only retaining the most highly expressed isoform per gene. For each k-mer length we calculated the difference in frequency obtained from transcripts with differing localization and/or region (e.g., the difference between cytosolic CDS and membrane CDS frequencies). To compare different k-mer lengths we next computed the z-scores of the differences in frequency for the top 40 HDLBP crosslinked k-mers (as described above) and all other k-mers. Z-score distributions were then compared between different k-mer lengths using pairwise Wilcoxon rank-sum tests.

To obtain multivalency scores for crosslinked four-mers we counted all crosslinked four-mers in the region +40/−40 nt around each T-C transition position. This was achieved with the findOverlaps function of the Genomic Ranges package (v.1.36.0, Bioconductor) to obtain the nucleotide distances between all T-C transitions within a transcript. The closest region (+4nt/−4nt) around the reference T-C position was excluded from counting since it would otherwise dominate the total crosslink signal within the +40/−40 nt regions. Four-mers were then ranked by the frequency of detected four-mers, which was obtained by dividing with the total number of all detected four-mers within the +40/−40 nt regions excluding the closest region. For the top ten enriched four-mers by frequency, we then binned the multivalency scores into five equally-sized categories and compared the total normalized T-C transition signal over the +40/−40 nt regions. Next, we calculated the percentage of total T-C transitions for every nucleotide position within the +40/−40 nt region for each multivalency bin.

Finally, we analyzed the multivalency potential of sequences of differentially localized transcripts and their regions. We chose two four-mer groups, namely the positive set (UUCU), which consisted of the top ten crosslinked four-mers, and the

negative set (AAGU), which had a similar transcriptome frequency as the positive set but was not enriched in HDLBP PAR-CLIP. Next, we counted the occurrence of both sets in 30-nt sliding windows in transcript sequences using vcountPDict function in Biostrings. We kept all windows with at least 3 four-mer group counts and summed them over transcript regions to obtain their frequency. The distribution of the frequencies was then compared between differentially localized transcripts and their regions with Wilcoxon rank-sum tests.

**Mapping PAR-CLIP crosslinks to ribosome and SRP 3D structures**. For this analysis we used our pre-rRNA and RN7SL alignments and extracted T-C transitions as described above in the PAR-CLIP section. Next, we aligned the pre-rRNA and RN7SL fasta files to the 18 S rRNA and RN7SL fasta files obtained from the published structures (PDB id's 4V6X, 6FEC, and 3JAJ)[41,89,90] and defined the orthologous position of the T-C transition. Next, we used Pymol (v.2.3.3) to simultaneously visualize initiation factors and expansion segments, as well as the SRP and expansion segments. Therefore, we juxtaposed 4V6X and 6FEC structures, as well as 4V6X and 3JAJ structures using the align command. Crosslinked nucleotides and other structural features were then labeled to obtain the final images presented in Figs. 4b and S4A.

**Protein expression and purification**. The pQLinkG expression plasmid encoding full-length Homo sapiens HDLBP as N-terminal GST-fusion protein was transformed in Escherichia coli Rosetta 2(DE3) (Merck KGaA). 8 L of LB medium, containing 100 μg/mL ampicillin and 43 μg/ml chloramphenicol were inoculated with 80 mL of an Escherichia coli Rosetta 2(DE3) overnight culture and grown in a Multitron HT incubator shaker (Infors) at 37 °C, 150 rpm until OD600 reached 0.7. At that point, the temperature was reduced to 18 °C, and protein expression was induced by the addition of 0.2 mM IPTG. The cultures were grown for a further 20 h. Cells were pelleted by centrifugation at $4000 \times g$ for 15 min. Pellets were resuspended in a buffer containing 50 mM Tris-HCl, pH 7.8, 500 mM NaCl, 4 mM MgCl2, 0.5 mM tris-(2-carboxyethyl)-phosphine (TCEP), mini-complete protease inhibitors (Merck KGaA, 1 tablet per 50 mL). 35 mL buffer per pellet from 1 L culture were used. About 5 μL of Benzonase (Merck KGaA) were added and the cells were lysed by passing the suspension at least twice through a Microfluidiser (Microfluidics). Lysates were clarified by centrifugation at $48,000 \times g$ for 45 min at 4 °C. Protein purification was performed at 4 °C on an Äkta pure FPLC (Cytiva) using an XK 16/20 chromatography column (Cytiva) containing 10 mL Glutathione Sepharose 4 Fast Flow beads (Cytiva). The column was washed with 250 mL of buffer containing 50 mM Tris-HCl, pH 7.8, 500 mM NaCl, 4 mM MgCl2, 0.5 mM TCEP, and eluted with 50 mL of buffer containing 50 mM Tris-HCl, pH 7.8, 500 mM NaCl, 4 mM MgCl2, 0.5 mM TCEP, 20 mM GSH. Eluent fractions were analyzed by SDS-PAGE, and appropriate fractions were pooled and reduced to 0.5 mL using centrifugal filter devices (Sartorius, Göttingen, Germany). As second purification step, gel filtration chromatography (GF) was performed at 4 °C on an Äkta prime plus FPLC (Cytiva). A Superdex 200 Increase 10/300 GL column (Cytiva) was equilibrated in 10 mM Tris-HCl pH 7.8, 150 mM NaCl, 4 mM MgCl2, 0.5 mM TCEP buffer, and the sample was eluted in the same buffer at a flow rate of 0.5 mL/min. Peak fractions were analyzed by SDS-PAGE, appropriate fractions were pooled and concentrated to ~5 mg/mL. The GST-tag was removed by incubation with TEV protease in a 1:30 (w/w) ratio for 16 h at 4 °C. To separate HDLBP from cleaved GST tag and the protease, a second Superdex 200 Increase 10/300 GL GF run was performed under the same conditions as the first run. Peak fractions were analyzed by SDS-PAGE, appropriate fractions were pooled, concentrated to ~5 mg/mL, flash-frozen in small aliquots in LN2, and stored at −80 °C.

The HDLBP protein variants A-D were produced at 17 °C using E. coli BL21-AI cells (Thermo Fisher Scientific) induced with 0.5 mM isopropyl β-D-1-thiogalactopyranoside (IPTG). For purification, cells were resuspended in lysis buffer (1× PBS pH 7.4, 0.2 M NaCl, 5% glycerol) supplemented with 0.25% (w/v) 3-[(3-cholamidopropyl)-dimethylammonio]−1-propanesulfonate (CHAPS), 1 mM phenylmethyl-sulfonyl fluoride (PMSF), 3000 U/mL lysozyme (Serva), and 7.5 U/mL RNase-free DNase I (AppliChem), lysed by multiple freeze-thaw cycles and the extract was cleared by centrifugation at $34,000 \times g$. The respective GST-fusion protein was captured from the supernatant using glutathione sepharose affinity chromatography on a GSTrap™ FF column (Cytiva) equilibrated with 20 mM Tris-HCl pH 8.0 and 0.2 M NaCl. The eluted protein was supplemented with 5 mM magnesium chloride and 1 mM DTT, and further purified by ion-exchange chromatography on a HiTrap Heparin HP column (Cytiva; for constructs B and D equilibrated with 1× PBS pH 7.4, 5% glycerol, 1 mM DTT; for construct A equilibrated with 20 mM phosphate buffer pH 6.0, 5% glycerol, 1 mM DTT) or on a 5 mL Source 30Q column (Cytiva; construct C; equilibrated with 10 mM sodium phosphate pH 7.4, 5% glycerol, 1 mM DTT), respectively. Purification of construct C additionally included a size-exclusion chromatography step on a Superdex 200 Increase 10/300 GL column (Cytiva) equilibrated with 1× PBS pH 7.4, 0.15 M NaCl, 5% glycerol, and 1 mM DTT. The purified proteins were flash-frozen with liquid nitrogen and stored at −80 °C until further use. The molecular mass of all purified constructs was confirmed by LC/MS TOF mass spectrometry.

**Fluorescence polarization assay**. RNA oligos were labeled with fluorescein in a two-step procedure as previously described[91]. Firstly, 1 nmol of RNA oligonucleotides was 5′-end thiophosphorylated overnight at 37 °C in 1× T4 polynucleotide kinase buffer supplemented with 0.5 mM ATP-γ-S, 5 mM DTT, and 10 units of T4 polynucleotide kinase followed by ethanol precipitation. Fluorescein was then added to the 5′ end of the RNA by incubating the RNA with 1 mM fluorescein maleimide for 2 h at room temperature in the dark in 50 mM phosphate buffer pH 7.0 followed by ethanol precipitation. The fluorescence polarization assay was performed as described before[92] with minor modifications. Briefly, serial dilutions of either full-length HDLBP or its various fragments with N-terminal GST-tag in 1× binding buffer (20 mM Tris, pH 7.5, 60 mM KCl, 1 mM EDTA, 10% glycerol, 1 ng/μl tRNA, 1 ng/μl heparin, 0.4 U/μl RNasin, and 200 ng/μl BSA) were first added to the wells of a black 384-well flat-bottomed microplate (Corning® NBS™) followed by addition of the fluorescein-labeled RNA probes to a final concentration of 10 nM. The final reactions were mixed by a microplate shaker, spun, and were incubated on ice for 20 min in the dark. The anisotropy values were measured and automatically calculated by the fluorescence polarization function of microplate reader SpectraMax iD5 (Molecular Devices), using the SoftMax® Pro 7 software.

**Live single-molecule imaging and ER co-localization analysis**. Imaging and image data analysis were performed analogously to the protocols described before[18]. In brief, HDLBP KO/Ctrls were generated in HeLa 11ht cell lines expressing Gaussia luciferase reporter transcripts from a single genomic location. Reporter transcript expression was induced by the addition of doxycycline for 1 h. Images were acquired 1–2 h after removal of doxycycline from the medium. The MS2-stem-loop encoding reporter transcripts were detected via MS2 coat protein-Halo fusion proteins that were labeled with Janelia dye JF549 and stably co-expressed from the HeLa cell genome. The ER was detected using a Turq2-KDEL ER marker that was also stably expressed in the cells. The image series consisted of 100 frames that were acquired at frame rates of 20 Hz (50 ms exposure). Single-particle mobility and ER co-localization analysis included all tracks obtained from single-particle tracking that were longer than 2 frames.

**Additional published datasets**. IRE1 PAR-CLIP[93] was obtained by communication with the corresponding author, SSB PAR-CLIP datasets[87,94] were obtained from GEO under accession "GSE95683" and from SRA under accession "SRR4301753". MOV10 PAR-CLIP[61] was obtained from GEO under accession "GSE48245".

**Reproducibility**. Western analyses presented in Figs. 4f, 5a, g and Supplementary Fig. S1a, d, were performed two times. SDS-PAGE analysis presented in Supplementary Fig. S3g was performed two times.

**Oligonucleotides**. Oligonucleotides are listed in Supplementary Table 1 in Supplementary Information.

**Reporting summary**. Further information on research design is available in the Nature Research Reporting Summary linked to this article.

## Data availability

The data supporting the findings of this study are available from the corresponding authors upon reasonable request. PAR-CLIP, RNA-seq, and ribosome profiling data from this study have been submitted to the NCBI Gene Expression Omnibus (GEO) under accession GSE148262.

BioID and pSILAC data have been submitted to the ProteomeXchange under the dataset identifiers "PXD018313 and PXD018316", respectively. Source data for the figures and supplementary figures are provided as a Source Data file.

## Code availability

All analysis scripts and processed data are publicly available from GitHub at https://github.com/mmilek/hdlbp_rev.git (https://doi.org/10.5281/zenodo.6347386).

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

## Acknowledgements

We are indebted to Aram Amalietti and Jernej Ule for the discussion of strategies for detecting multivalency in CLIP data. We thank Ouidad and Nouhad Benlasfer for the generation of stable cell lines and RNA-seq library preparation. We thank Anja Schütz and the team of the Protein Production & Characterization technology platform of the Max Delbrück Center for Molecular Medicine, Berlin, Germany, for producing the various HDLBP constructs (A-E). We acknowledge Madlen Sohn, Kirsten Richter, and Sascha Sauer from the Genomics technology platform of the Max Delbrück Center for Molecular Medicine for high-throughput sequencing. We thank members of the Landthaler laboratory for fruitful discussions. The Jeff Chao laboratory is supported by the Novartis Research Foundation. David Schwefel is supported by the DFG Emmy Noether program SCHW 1851/1-1.

## Author contributions

U.Z., M.M., and M.L. conceived the study and designed experiments. U.Z. performed the majority of the experimental work, analyzed data, and prepared figures. M.M. carried out the majority of the computational data analysis and prepared the majority of figures including Supplemental material. I.M. conducted FP assays and was supervised by M.L. C.H.V.-V. contributed pSILAC mass spectrometry experiments and was supervised by M.S. G.M. and S.K. contributed to BioID mass spectrometry. N.B. carried out flow cytometry experiments and S.M. carried out phenotyping and mouse xenograft experiments both under the supervision of S.H. F.V. contributed luciferase mRNA localization data and were supervised by J.A.C. O.-G.H. contributed computational data analysis. D.S. produced and purified full-length HDLBP. S.D.G. contributed validation experiments. U.Z., M.M., and M.L. interpreted data with input from all authors and wrote the manuscript with contributions from all authors.

## Funding

## Competing interests

The authors, M.M., C.H.V.-V., S.M., G.M., O.-G.H., S.D.G., D.S., N.B., F.V., J.A.C., S.K., S.H., and M.S. declare no competing interests. The authors U.Z., I.M., and M.L. declare the following competing interests. U.Z., I.M., and M.L. are co-inventors on a provisional patent application filed by the Max-Delbrück Center for Molecular Medicine related to results presented in Fig. 5.
