## [Peer Review File · Nature Communications]

Title: HDLBP binds ER-targeted mRNAs by multivalent interactions to promote protein synthesis of transmembrane and secreted proteinsEditorial Note: This manuscript has been previously reviewed at another journal that is not operating a transparent peer review scheme. This document only contains reviewer comments and rebuttal letters for versions considered at *Nature Communications*.

REVIEWER COMMENTS

Reviewer #2 (Remarks to the Author):

In the revised version of the manuscript, the authors have responded thoroughly to all concerns raised in the initial review and substantial new data are included where seeming deficiencies in the original manuscript were noted. The author's response and rebuttal is thorough, well argued, and again supported by additional data. En toto, this is a tour de force study and although there is substantial overlap with the prior Mobin et al. 2016 *Nature Communications* 7, 12848 report, as noted in my prior review, the manuscript in its current state is indeed a substantial advance and remarkable in the depth and comprehensive analysis of HDLBP function in ER-localized translation. Indeed, with the growing interest in the field, this study provides the most compelling and complete analysis of RNA binding protein function in ER-localized translation to date. My recommendation is that the manuscript be accepted and published without further delay.

Reviewer #3 (Remarks to the Author):

I was recruited to review this revised manuscript, due to the unavailability of the original reviewer #1.

This manuscript provides a deeper analysis of the RNA-binding protein HDLBP / Vigilin. Fluorescence anisotropy data in the current version support the conclusion that this protein binds C/U-rich regions and that longer C/U tracts show better binding. CLIP data indicate that the protein binds principally to the coding sequences of mRNAs encoding secreted proteins, with some binding also seen on the 3' UTRs of cytosolic transcripts. HDLBP also crosslinks to the ribosomal RNA, suggesting a direct interaction. Loss of HDLBP reduces ribosome occupancy on target transcripts and is argued to shift the ribosome occupancy on specific codons, especially those matching the HDLBP binding preference. Loss of HDLBP also impairs many aspects of A549 cell growth in culture.

I share the sense that the work provides a broader and higher resolution view of HDLBP function, but does not truly change our understanding of its mechanism or view of its role in the cell. The revisions have addressed many of the key concerns with the original submission raised by reviewer #1, and the authors responses have clarified other points.

1. I remain concerned about point 4 raised by reviewer #1. The analysis presented in Figure 3E, and discussed in the response, seems a bit confounded to me. A "highly multivalent" region has multiple

individual C/U binding motifs in a 40 nt window, and independent binding at each of these motifs should lead to more T-C conversions in the window. Such a region might show more T-C conversions because it has more extensive contacts when it is occupied, but no fractional occupancy and thus no higher affinity. Furthermore, crosslinks are only detected at Us in the RNA, which are correlated with these motifs.

While the fluorescence anisotropy data confirm the C/U tracts are bound with higher affinity than C/A tracts, they do not test cooperativity between motifs on the same oligo or the proposed multivalency.

I also have concerns with the interpretation of the ribosome footprint profiling data. These were not raised in the initial review, but they are important discussion points that could be addressed by textual changes.

2. Ribosome profiling directly reports on ribosome occupancy, which can be affected by changes in the speed of elongation as well as changes in the frequency of initiation. It thus seems possible that the ribosome profiling changes seen on HDLBP targets arise due to faster elongation — that is, loss of HDLBP alters ribosome pausing, but in the opposite manner proposed by the authors.

3. Changes in P- and E-site occupancy (Fig 6A) appear modest and I suspect are not significant after multiple testing correction.

4. These changes are also hard to understand mechanistically. It is unclear to me how HDLBP could either promote, or stall, ribosomes translating exactly the codons bound by HDLBP. Roughly 12 bases of mRNA are protected by the ribosome upstream of the A-site codon, and it seems that other RNA-binding proteins are necessarily removed when they meet the leading edge of the ribosome.

5. As a related point, is it hypothesized that HDLBP quickly rebinds mRNAs after the ribosome passes? It is often proposed that coding sequences have low occupancy by RNA-binding proteins because translation strips these proteins off even when they could bind the RNA with higher affinity.

6. On p. 18, it is remarked that “this dataset was obtained from non-fractionated cells” and then, “[a]s expected we observed low footprint density for SP and TM containing mRNAs in the region downstream of the start codon, until the emergence of both targeting signals from the ribosome” I’m not aware of this as a general or expected feature of total (non-fractionated) ribosome footprints on secreted proteins.

We would like to thank the reviewers for their very constructive criticism and very insightful comments on our work. We were delighted to read that the reviewers recognized the value of our findings:

Below we provide our point-by-point response to the reviewers' comments (in red).

Reviewer #2:

In the revised version of the manuscript, the authors have responded thoroughly to all concerns raised in the initial review and substantial new data are included where seeming deficiencies in the original manuscript were noted. The author's response and rebuttal is thorough, well argued, and again supported by additional data. En toto, this is a tour de force study and although there is substantial overlap with the prior Mobin et al. 2016 Nature Communications 7, 12848 report, as noted in my prior review, the manuscript in its current state is indeed a substantial advance and remarkable in the depth and comprehensive analysis of HDLBP function in ER-localized translation. Indeed, with the growing interest in the field, this study provides the most compelling and complete analysis of RNA binding protein function in ER-localized translation to date. **My recommendation is that the manuscript be accepted and published without further delay.**

We are delighted to have addressed the reviewer's concerns.

Reviewer #3 (replacing Reviewer #1):

I was recruited to review this revised manuscript, due to the unavailability of the original reviewer #1.

This manuscript provides a deeper analysis of the RNA-binding protein HDLBP / Vigilin. Fluorescence anisotropy data in the current version support the conclusion that this protein binds C/U-rich regions and that longer C/U tracts show better binding. CLIP data indicate that the protein binds principally to the coding sequences of mRNAs encoding secreted proteins, with some binding also seen on the 3' UTRs of cytosolic transcripts. HDLBP also crosslinks to the ribosomal RNA, suggesting a direct interaction. Loss of HDLBP reduces ribosome occupancy on target transcripts and is argued to shift the ribosome occupancy on specific codons, especially those matching the HDLBP binding preference. Loss of HDLBP also impairs many aspects of A549 cell growth in culture.

I share the sense that the work provides a broader and higher resolution view of HDLBP function, but does not truly change our understanding of its mechanism or view of its role in the cell. The revisions have addressed many of the key concerns with the original submission raised by reviewer #1, and the author's responses have clarified other points.

1. I remain concerned about point 4 raised by reviewer #1. The analysis presented in Figure 3E, and discussed in the response, seems a bit confounded to me. A “highly multivalent” region has multiple individual C/U binding motifs in a 40 nt window, and independent binding at each of these motifs should lead to more T-C conversions in the window. Such a region might show more T-C conversions because it has more extensive contacts when it is occupied, but no fractional occupancy and thus no higher affinity. Furthermore, crosslinks are only detected at Us in the RNA, which are correlated with these motifs.

While the fluorescence anisotropy data confirm the C/U tracts are bound with higher affinity than C/A tracts, they do not test cooperativity between motifs on the same oligo or the proposed multivalency.

The original point 4 raised by reviewer #1 questioned the presence of a high number of T-C transitions in the highly multivalent regions around the central crosslinks. We have indeed addressed this point in Fig. 3E, where we stratified the 40nt regions based on the multivalency score and plotted the total number of T-C transitions between different regions. It follows from this analysis that highly multivalent regions also show the highest T-C transitions, which suggest the highest affinity of HDLBP towards multivalent regions. However, as pointed out by the reviewer, high number of HDLBP-bound motifs and/or Us in these regions could also lead to a higher number of T-C transitions even if affinity would not be higher.

To address this concern, we now present direct affinity measurements from fluorescence anisotropy binding experiments of HDLBP to synthetic RNA molecules with different numbers of bound 4-mer motifs within a longer (34nt) sequence (Fig.3I and below). We determined significantly lower K_d values for synthetic RNA molecules containing 3 HDLBP-bound 4-mers (CUUC or UCUU) than those harboring only 2 motifs. Therefore, HDLBP (KH5-9) has approximately 3-5-fold higher affinity for RNAs with higher multivalent potential.

RNA oligonucleotides with different multivalency potential

- H40 AAAAAACUUCAAAAAAAAAAAAAAAAACUUCAAAAAA
- H41 AAAAAACUUCAAAAACUUCAAAAACUUCAAAAAA
- ▲ H44 AAAAAACUUCAAAAACUUCAAAAACUUCAAAAAA
- ▲ H42 AAAAAUCUUAAAAAAAAAAAAAAAAUCUUAAAAAA
- ▲ H43 AAAAAUCUUAAAAUCUUAAAAUCUUAAAAAA

These findings strongly support our conclusions from PAR-CLIP analysis and confirm that HDLBP binds with high affinity to long RNA regions via multivalent interactions.

Reviewer #3: I also have concerns with the interpretation of the ribosome footprint profiling data. These were not raised in the initial review, but they are important discussion points that could be addressed by textual changes.

2. Ribosome profiling directly reports on ribosome occupancy, which can be affected by changes in the speed of elongation as well as changes in the frequency of initiation. It thus seems possible that the ribosome profiling changes seen on HDLBP targets arise due to faster elongation — that is, loss of HDLBP alters ribosome pausing, but in the opposite manner proposed by the authors.

We thank the reviewer for this comment and agree that mechanistic effects of HDLBP could (1) involve promotion of elongation arrest upon translation of the targeting signals or could (2) contribute to global elongation fidelity. Below, we summarize our findings that support but do not fully answer these questions:

Fig. 5H: The local footprint coverage in the WT condition immediately downstream of the targeting signals is reproducibly higher than in the KO condition, suggesting that HDLBP contributes to elongation arrest that is required for localization to the ER membrane. Decreased translation efficiency in the HDLBP KO conditions may be a consequence of reduced local elongation arrest at targeting signals, otherwise required for efficient translation elongation¹.

Fig. 6A: The global codon shifts in the E-site calculated from entire coding sequences show that the stalling in HDLBP-bound and other codons are modestly increased in the absence of HDLBP. This may suggest that HDLBP prevents global stalling and promotes elongation. However, since

these shifts are modest and not statistically significant, they may be a consequence of indirect effects of HDLBP KO.

Since the mechanism of HDLBP effects are unclear, we have now modified our conclusions in the results and discussion sections (see pages 23, 31) to be in line with the data presented.

3. Changes in P- and E-site occupancy (Fig 6A) appear modest and I suspect are not significant after multiple testing correction.

We redid the P- and E- site occupancy analysis using ribosome profiling data with approximately 5-fold higher coverage. The codon shifts have remained almost the same as initially presented. Even though there were only two observations per group (two WT and two KO replicates for each guide RNA), we also performed multiple testing corrections as suggested by the reviewer and we now present these results below.

Based on this additional analysis, we conclude that the global changes in P-site and E-site occupancy are modest and not statistically significant. Therefore, they may reflect indirect effects of HDLBP. We amended the main text accordingly and added a sentence in the discussion.

P-site

E-site

4. These changes are also hard to understand mechanistically. It is unclear to me how HDLBP could either promote, or stall, ribosomes translating exactly the codons bound by HDLBP. Roughly 12 bases of mRNA are protected by the ribosome upstream of the A-site codon, and it seems that other RNA-binding proteins are necessarily removed when they meet the leading edge of the ribosome.

We thank the reviewer for this comment. We have not addressed the positioning of HDLBP crosslinked motifs containing the codons relative to the position of the stalled codons. It may well be that the relevant HDLBP crosslinks appear up- or downstream rather than exactly at the stalled

codon. Since stalling at specific codons was only modestly present and not statistically significant, we are not confident in performing this analysis or derive any conclusions from it.

In addition, we believe that it would be extremely difficult to perform meaningful analysis of codon stalling relative to the position of HDLBP binding sites. Due to the multivalent binding of HDLBP, which is present over long RNA regions, it would be very difficult to define the main binding position in order to be able to assess the distance to the stalled codon.

Nevertheless, we modified a sentence in the discussion to speculate that HDLBP is bound to the mRNA downstream from the ribosome, resulting in collisions with the elongating ribosome (page 31). The sentence now reads:

Since HDLBP is most likely bound to the mRNA downstream from the elongating ribosome, it could be sequestered from the mRNA via ribosome collisions and/or possibly via tRNA and/or ribosome ES6SB-dependent mechanism, which remains to be elucidated.

5. As a related point, is it hypothesized that HDLBP quickly rebinds mRNAs after the ribosome passes? It is often proposed that coding sequences have low occupancy by RNA-binding proteins because translation strips these proteins off even when they could bind the RNA with higher affinity.

We thank the reviewer for this comment and agree that this is an important question. As stated in the discussion, we speculate that HDLBP is bound to the mRNA only during the primary round of translation after which it is removed. However, we have no data at this time that would allow us to understand the dynamics of HDLBP binding during active translation.

In addition, the high-affinity binding site at ES6SB determined by PAR-CLIP could function in HDLBP recruitment to or sequestration from actively translated mRNA. Similarly, many tRNA interactions may also be a consequence of dynamic mRNP assembly or disassembly but potentially also to prevent diffusion of discharged tRNAs and keep the rRNA pool high, as previously suggested². At present we are unfortunately unable to delineate the succession of HDLBP binding events during different stages of translation, as we have also indicated in the discussion.

6. On p. 18, it is remarked that “this dataset was obtained from non-fractionated cells” and then, “[a]s expected we observed low footprint density for SP and TM containing mRNAs in the region downstream of the start codon, until the emergence of both targeting signals from the ribosome”

I'm not aware of this as a general or expected feature of total (non-fractionated) ribosome footprints on secreted proteins.

We thank the reviewer for pointing this out. The ribosome footprint profiles of membrane-bound and cytosolic mRNAs should indeed differ, since ER-translated mRNAs are only localized to the membrane fraction after the targeting signal is translated in the cytosol. If membrane localized translation is captured, the average footprint profile of ER-targeted mRNAs should show lower coverage in the region immediately downstream of the start codon where the signal peptide or the 1st TMD are located than in the case of transcripts translated in the cytosol.

Our concern was that the dataset obtained from non-fractionated cells contained mostly footprints that were derived from the cytosolic ribosome. In that case, we would observe P-site coverage to show no decrease in the initial nucleotides after the start codon for both cytosolic and membrane-localized transcripts. However, our analysis demonstrated that membrane-bound ribosomes were also captured in the data, due to local differences in the footprint coverage, as shown in Supplemental Figure S5C.

References

1. Lakshminarayan R, *et al.* Pre-emptive Quality Control of a Misfolded Membrane Protein by Ribosome-Driven Effects. *Curr Biol* **30**, 854-864 e855 (2020).
2. Hirschmann WD, Westendorf H, Mayer A, Cannarozzi G, Cramer P, Jansen RP. Scp160p is required for translational efficiency of codon-optimized mRNAs in yeast. *Nucleic Acids Res* **42**, 4043-4055 (2014).

REVIEWERS' COMMENTS

Reviewer #3 (Remarks to the Author):

The revised manuscript has addressed my concerns with the earlier version. The new analysis of synthetic 2x and 3x motif RNAs supports the model for multivalent HDLBP binding; I would suggest changing the colors in the bar graph to match the line plot.

Point-by-point Response:

Reviewer #3 (Remarks to the Author):

The revised manuscript has addressed my concerns with the earlier version. The new analysis of synthetic 2x and 3x motif RNAs supports the model for multivalent HDLBP binding; I would suggest changing the colors in the bar graph to match the line plot.

Colors in Figure 3i were changed.